# A Decomposition-Based Multi-Objective Evolutionary Algorithm for Solving Low-Carbon Scheduling of Ship Segment Painting

Henan Bu [1,*], Xianpeng Zhu [1], Zikang Ge [1], Teng Yang [1], Zhuwen Yan [2] and Yingxin Tang [2]

[1] School of Mechanical Engineering, Jiangsu University of Science and Technology, Zhenjiang 212100, China; zxp1875477582@163.com (X.Z.); 17856006090@163.com (Z.G.); 18296956041@163.com (T.Y.)
[2] Industrial Technology Research Institute of Intelligent Equipment, Nanjing Institute of Technology, Nanjing 211167, China; zhuwen19880301@163.com (Z.Y.); njtangyingxin@outlook.com (Y.T.)
* Correspondence: buhn_just_edu@163.com

**Abstract:** Ship painting, as one of the three pillars of the shipping industry, runs through the whole process of ship construction. However, there are low scheduling efficiency and excessive carbon emissions in the segmental painting process, and optimizing the scheduling method is an important means to achieve the sustainable development of the ship manufacturing industry. To this end, firstly, a low-carbon scheduling mathematical model for the segmented painting workshop is proposed, aiming to reduce carbon emissions and improve the painting efficiency of the segmented painting workshop. Second, an artificial bee colony algorithm designed based on a decomposition strategy (MD/ABC) is proposed to solve the model. In the first stage, five neighborhood switching methods are designed to achieve the global search employed for each solution. In the second stage, the Technique of Ordering the Ideal Solutions (TOPSIS) improves the competition mechanism through the co-evolution between neighboring subproblems and designs the angle to define the relationship between neighboring subproblems to enhance the localized search and improve population quality. The solution exchange strategy is used in the third stage to improve the efficiency of the algorithm. In addition, a two-stage coding method is designed according to the characteristics of the scheduling problem. Finally, the algorithm before and after the improvement and with other algorithms is analyzed using comparative numerical experiments. The experimental results show the effectiveness of the algorithm in solving the low-carbon scheduling problem of ship segmental painting and can provide reliable guidance for the scheduling program of segmented painting workshops in shipyards.

**Keywords:** ship segment painting; green scheduling; multi-objective optimization; decomposition





## 1. Introduction

With the implementation of the carbon peak and carbon neutral "double carbon" strategy, energy saving and emission reduction are an important challenge for the manufacturing industry, which has increasingly been the concern of the majority of countries, and green painting of ships is a new idea for low-carbon development [1,2]. Ship painting is a crucial process in ship construction and is also the most important source of pollution, emitting volatile organic compounds (VOCs), which not only harm the environment but also affect human health. According to statistics, the manufacturing industry accounts for about half of the global energy use and 38% of the global carbon dioxide emissions, while VOC pollution control will cause the rise of energy consumption of equipment operation, which is one of the important sources of carbon emissions in the ship manufacturing industry [3,4]. Therefore, reducing carbon emissions from ship painting has become a top priority. Segment painting is the most important and basic part of ship painting work; except for the special parts of special ships, all parts of the hull are partially or completely coated at this stage. The process of sectional painting includes several main steps. Firstly,

the structural integrity of the segments must be checked before painting, and then, the surface treatment and painting of the segments are carried out. After the segmental painting is completed, the painting inspection is carried out when the painting is sufficiently dry, mainly checking whether the painting coverage is complete, whether the painting thickness is up to standard, and whether the paint is uniform. In addition, the majority of domestic shipyards' segmented painting workshops are equipped with sandblasting guns and high-pressure airless spraying equipment with a low degree of automation, and workers consume a lot of energy during the operation. As the main energy-consuming link in ship painting, the segmented painting workshop needs to implement energy-saving and emission-reduction optimization policies. Scientific and reasonable low-carbon scheduling for painting plays a key role in the ship segment painting process, which helps optimize the painting production process, improve transportation efficiency, reduce paint consumption, and, at the same time, minimize energy consumption for VOC exhaust gas treatment, thus reducing carbon emissions.

At present, most of the sectional painting in China's shipyards is in the process of painting scheduling along the rough scheduling mode; that is, by virtue of the experience of the field staff and the traditional scheduling model to develop a plan, issued to the painting workshop for production, did not form scientific sectional painting scheduling processes and norms. Lee et al. studied the segment scheduling problem in the production planning of a large shipbuilding enterprise, established a hybrid integer linear planning model about region allocation and assembly shop scheduling, aiming to minimize the sum of the assembly shop's workload deviations in a specific period, and proposed a two-stage algorithm to solve the model in a finite period of time, and the experimental results showed that the algorithm exhibits superior performance in solving the shipyard data instances [5]. Cho et al. studied the spatial scheduling problem of a ship painting operation, which aimed to consider the allocation of painting workshop and space and balance the workload of the painting team, propose an algorithm of spatial scheduling, and develop a corresponding system. The experiment proved that the system solves the space scheduling problem of ship segmental painting well and balances the workload of the painting team [6]. It can be seen that the existing scheduling methods in ship construction face many challenges, such as the underutilization of time, space, and resources. It is worth noting that shipping companies are facing a green transition due to the huge increase in carbon emissions and energy consumption costs [7–9]. Therefore, adopting low-carbon concepts and intelligent algorithms to realize low-carbon scheduling of ship segment painting workshops is of great significance for enhancing China's ship manufacturing industry and is also an important way to promote the development of intelligent and green ship painting.

Researchers have conducted a large number of studies to address the low-carbon scheduling problem. For example, Xiang et al. developed a scheduling model based on carbon capture, carbon emissions trading demand response, and renewable energy generation to solve the low-carbon economic scheduling problem of the power-gas system, studied the economic and carbon emission benefits under the synergy of different low-carbon technologies, balanced the economy, carbon emission, and risk by setting the confidence level reasonably, and experimentally proved the feasibility and effectiveness of the proposed model by applying the case study to the proposed model [10]. Wu et al. aimed to reduce unnecessary energy consumption in the industrial sector by investigating the shutdown mechanism of whether and when to turn off the power supply and set different speed levels and proposed a green scheduling algorithm to solve the problem, and the case results validated the performance of the algorithm [11]. M. Geetha et al. considered the furniture manufacturing industry's carbon scheduling problem and developed a hybrid optimization algorithm with sequential hybridization and experimentally analyzed the strategies introduced by this algorithm, and the results show that it can reduce the carbon footprint by 9.82% [12]. In summary, scholars have considered the energy consumption and carbon emissions of various industries, such as electric power and furniture manufacturing, at the levels of model constraints, machine optimization, and scheduling strategies. However,

there are few reports in the field of ship construction involving low-carbon scheduling for painting.

The high demand for low-carbon scheduling is highlighted by the complexity of the ship segmental painting process and the varying levels of productivity and energy consumption generated between painting operators and equipment. The low-carbon scheduling of ship-segmented painting considers segmental sequencing along with the selection of the painting team, and the complexity of the problem is more complicated than the classical hybrid flow-shop scheduling problem (HFSP). Since the past research on HFSP mainly focuses on a single objective function and does not take into account the real problems, research on multi-objective problems (MOPs) has been increasing year by year [13,14]. Qiu et al. systematically examined and evaluated a series of clustering algorithms and models in order to solve the problem of accurate customer segmentation for the financial card industry, and the experimental results showed that the method could deepen the segmentation of customers and promote economic growth [15]. Heydarpoor et al. provided a proper medical supervision model for optimizing tumor treatment. The model was developed to minimize the cancer cell density and the number of approved drugs and solved using a meta-search algorithm, and the experimental results show that the algorithm has superior performance in terms of convergence and scalability [16]. Homaee et al. investigated the effect of the diameter of the refill valve on pre-strike during the closing process of a pressurized gas-type sulfur hexafluoride circuit breaker to establish a multi-physics field model, and numerical studies showed that the diameter of the re-fill valve has a great influence on the moment of pre-strike occurrence [17]. Luan et al. established a low-carbon scheduling model to minimize the workshop completion time and energy cost considering sustainable development and proposed a biologically based heuristic algorithm [18]. Wang et al. studied the energy efficient HFSP problem under machine failure, established a relevant mathematical model, and proposed an improved multi-objective firefly algorithm to optimize it [19]. However, there may be conflicting relationships between multiple objectives, and a balance must be sought between them.

Recent studies have shown that the decomposition-based multi-objective evolutionary algorithm (MOEA/D) shows better performance in solving MOP scheduling problems; the idea is to transform the MOP into multiple single-objective problems and use an evolutionary algorithm to optimize these single-objective problems at the same time [20–22]. Xiang et al. proposed a new algorithm combining the decomposition-based algorithm and the artificial bee colony (ABC) algorithm, which can take advantage of the advantages of both algorithms and maintain good diversity and convergence speed [23]. Zhang et al. proposed a decomposition-based three-stage multi-objective optimization algorithm for solving energy-efficient HFSPs that strikes a balance between local exploitation and global exploration [24]. Li et al. proposed a unified framework based on decomposition methods and incorporating dominance relations. By using weights to formulate several subregions of the target space and defining the minimum fitness value in the subproblems to improve the diversity of the population, the algorithm adopts a multi-layer selection approach, allowing the solution scheme to survive to the next round, and compared with other algorithms, the results show that the algorithm can balance the diversity and convergence in the evolutionary process [25].

Given the above deficiencies, this paper first establishes a scheduling model aiming at minimizing the completion time and total carbon emissions for the environment of ship-segmented painting workshops. A green scheduling algorithm for segmental painting, which consists of three stages, is proposed. In the first stage, five variable neighborhood structures are designed to optimize each scalar problem independently; in the second stage, the TOPSIS technique is used to utilize the more promising solutions, and a competitive mechanism is proposed for the superior solutions to replace the inferior ones to improve the quality of the population. In addition, the weight vector and its corresponding solutions are defined by restricting the angle between the improvement regions to ensure population diversity and convergence. In the third stage, a solution strategy based on segment ex-

change is designed to replace the solutions that fall into the local optimum to enhance the global exploration capability of the algorithm. Finally, the proposed algorithm is validated by process data of different numbers of segments, segmental painting workshop size, and dispatch orders in a shipyard in Shanghai and is compared to other algorithms, and the results show that the algorithm not only performs excellently in reducing the carbon emission of the painting workshop schedule but also can provide reliable guidance for the scheduling scheme of segmented painting workshop in shipyards. The contribution of this paper is as follows:

1. At this stage, ship enterprises are facing green transformation, and a ship segmental painting low-carbon scheduling model is established for the ship segmental painting low-carbon scheduling during the operation of the painting team during the ship segmental painting process, which is characterized by the problems of low efficiency and excessive carbon emission;
2. An MD/ABC algorithm is proposed for solving the above model.

The innovations of the MD/ABC algorithm are as follows:

(1) Designing a two-level coding method for ship segmental painting low-carbon scheduling model;
(2) Designing five neighborhood switching methods to ensure that the subproblems can be fully optimized and enhance the global exploration performance of the algorithm;
(3) Improving the competition mechanism by using TOPSIS technology and introducing an angle strategy to further enhance the local search capability of the algorithm;
(4) Designing an exchange strategy for the solutions of subproblems in different neighborhoods to further enhance the performance of the algorithm.

The rest of this paper is structured as follows: In Section 2, the process flow of ship segmental painting is analyzed, and a ship segmental painting low-carbon scheduling model is proposed. In Section 3, some strategies are introduced in this paper. In Section 4, the MD/ABC algorithm is proposed to solve the above model. The experimental results as well as the analysis are reported in Section 5. Finally, conclusions are given in Section 6.

## 2. Modeling of Low-Carbon Scheduling Problem for Ship Segment Painting

### 2.1. Segmental Painting Process

The production process of segmentation begins with the numbering of materials and passes through the multiple stages and production processes of segment assembly. In the construction stage, materials are transformed into hull parts, parts into components, and components into segments. After the assembly department completes the assembly work from raw materials to segments, the segments will be transferred to the painting department for segmental painting. The production process of segmental painting is roughly divided into two stages, sandblasting and painting, and the specific process is shown in Figure 1.

The sandblasting process in the segmented painting workshop is as follows:

- Segmentation into the sandblasting room: Large flatbed trucks are used to transport the sand to be blasted into the sandblasting room in segments and close the flexible door of the room;
- Safety inspection: It is performed mainly using manual methods, the comprehensive inspection of scaffolding, etc., to ensure the safety of construction personnel and inspection of construction tools and labor protection supplies;
- Equipment: It mainly includes indoor lighting, a whole room dust removal system, a dehumidification system, and a sandblasting system;
- Sandblasting operation: A sandblasting gun is used to treat the segments and remove the oxidized skin and other impurities on the surface of the steel plate to make it show metallic luster. At the same time, sufficient lighting is ensured to guarantee the safety of the operating personnel, and equipment such as dust removal and dehumidification are activated to control the temperature and humidity of the air and the concentration of dust in the plant;

- Sand collecting operation: At present, manual sand collecting and vacuum sand sucking machines are used in combination. The vacuum sand-sucking machine consumes a lot of electricity when used, and at the same time, it needs the whole room's dust removal equipment, dehumidifier, and other equipment to cooperate;
- Sectional cleaning and welding structural defects: They are performed mainly through manual pneumatic tools to remove welding spatters, free-cutting edges, running holes, and other structural defects, as well as residual dust on the surface of the steel plate. In order to meet the requirements and improve the quality of subsequent painting construction, it is necessary to cooperate with the whole room dust removal, dehumidifier and other equipment;
- Blasting internal and external inspection: After completing the sand blasting treatment, the segments need to be transported by flatbed trucks to the painting plant for inspection.

  The painting process in the segmented painting workshop is as follows:

- Spraying construction: The process of manually spraying paint on the surface of steel plates utilizing high-pressure airless spray pumps. The construction requires a small amount of compressed air, and dehumidifiers and organic solvent purification devices are enabled in the plant to ensure that harmful gases, such as VOC, are fully absorbed and environmentally friendly emissions are realized;
- Manual repair and pre-coating: The process of repairing areas that could not be effectively sprayed or were missed during the spraying process, usually by hand brushing or roller coating;
- Paint recoating: Recoating is performed using manual airless spraying construction;
- Paint repair and internal inspection: The film thickness test and the malpractice inspection of completed paint are employed to eliminate problems and ensure that the film-forming effect after construction meets the process requirements;
- External inspection of paint completion: It is performed mainly by the shipowner, paint service provider, crew construction personnel, etc., according to the construction process specifications and quality standards, etc., on the construction of the end of the segment for the external inspection of the film formation situation.

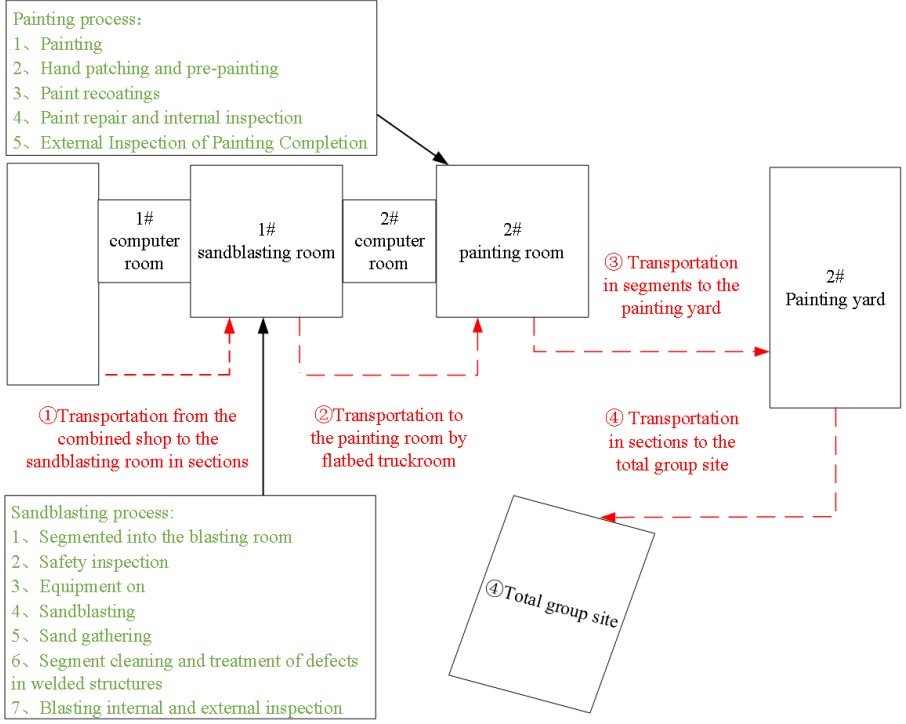

**Figure 1.** Segmental painting process flowchart.

## 2.2. Description of the Problem

The ship segmental painting low-carbon scheduling problem studied in this paper can be described as follows: There exists a group of segments that arrive at the sandblasting shop and have to pass through the various stages of the sandblasting shop continuously. Paint teams (consisting of equipment and operators) with different processing capabilities exist for each stage. Once the process is completed in the sandblasting shop, the segments are transported by flatbed trucks to the paint shop for painting. The painting team consumes a lot of energy during its operation. Different time and energy consumptions may be required for different segments. Between particular stages, the transportation distances of the segments are different and require different transportation times and energy consumption. The objective is to determine the sequence of segments for each stage and the allocation of painting teams to minimize the time to completion ($MS$) and the total carbon emissions ($EC$) to achieve an optimal mix that minimizes the carbon emissions generated by the energy consumption of the equipment. The following assumptions are presented:

- Each segment must pass through all phases sequentially, with each segment job assigned to a transportation device at a given phase;
- At any given time, a maximum of one segmental painting team may be processed, and a maximum of one segment may be processed on one painting team;
- Interruptions and preemption are not permitted;
- Sections are transported to the paint shop immediately after completion in the sandblasting shop;
- The buffer between the stages of the segmented painting workshop is unrestricted;
- The types of equipment turn on when the paint team is assigned the first segment to begin processing. Only when all segments of the paint team are completed do the types of equipment turn off;
- Segmented jobs must be completely processed by the painting team in the last stage before the next stage can be started;
- Paint teams are permitted idle time, and segments are not to be processed by the paint team until the adjacent operation has been adjusted;

The relevant symbols are listed in Table 1.

**Table 1.** Parameters and variables.

| Symbol | Meaning |
|---|---|
| $J$ | Set of segments and $J = \{1, \ldots, j, \ldots, n\}$ |
| $I$ | Set of stages and $I = \{1, \ldots, i, \ldots, s\}$ |
| $M_i$ | Set of painting teams at the stage $i$ and $M_i = \{1, \ldots, m, \ldots |M_i|\}$ |
| $L$ | A grouping of positions designed to accommodate segments allocated to individual painting teams and $L = \{1, \ldots, q, \ldots, n\}$ |
| $p_{i,j}$ | Segment $j$ Processing time for the segmental painting team on the stage $i$ |
| $t_{j,j',i}$ | Setup time from segment $j$ to $j'$ stage $i$ $j = j'$ represents that segment $j$ is the first segment assigned to a painting team |
| $f_{m,m',i}$ | Transportation from painting team $m$ on stage $i$ to paint team $m'$ on stage $i+1$ |
| $T^S_{j,j',i}$ | Setup energy consumption from segment $j$ to segment $j'$ in stage $i$. $j = j'$ indicates that segment $j$ is the first segmental painting team assigned at stage $i$ |
| $P^W_i$ | Energy consumption of the painting team in process on stage $i$ |
| $P^I_i$ | Energy consumption of the painting team at idle on stage $i$ |
| $P^F_i$ | Energy consumption of transportation equipment between $i$ and $i+1$ in a given phase |
| $v_{m,i}$ | Equipment energy utilization for the painting team $m$ on stage $i$ |
| $b_{i,j}$ | Start time for the segment $j$ during the stage $i$ |
| $e_{i,j}$ | Ending time for the segment $j$ during the stage $i$ |
| $B_{i,m,q}$ | The outset time for the segment at the position $q$ within painting team $m$ |

**Table 1.** *Cont.*

| Symbol | Meaning |
|--------|---------|
| $E_{i,m,q}$ | The finish time for the segment at the position $q$ within painting team $m$ |
| $x_{i,j,m,q}$ | The binary variable equals 1 when segment $j$ is allocated to position $q$ of painting team $m$ during stage $i$, and 0 otherwise |
| $z_{i,j}$ | The binary variable equals 1 if segment $j$ necessitates transportation at stage $i$; otherwise, it equals 0 |
| $Y_{i,m,q}$ | The intermediate variable denotes the energy consumption of the painting team $m$ on stage $i$ when it stays in the adjusted and idle state from position $q$ to position $q+1$, and $Y_{i,m,1}$ denotes the initial adjusted energy consumption of the first segmental painting team $m$ assigned to it |
| $W_{i,j}$ | The intermediate variable indicates the energy consumption to transport segment $j$ from a particular stage $i$ to stage $i+1$ |

*2.3. Carbon Emission Category Analysis*

Carbon emissions generated by the ship-segmented painting workshop mainly originate from the consumption of electric energy during the processing, including equipment (single-cylinder double-gun sand blasting machine, single-cylinder single-gun sand blasting machine, split-type vacuum sand suction machine, ground-type combined sand suction machine, flexible gates, four-season dehumidifier, catalytic combustion device for organic exhaust gases, combined air-handling unit, immersed cartridge dust collector, high-pressure airless spraying pumps, metal halide lamps, Light-Emitting Diode lamps, etc.) and other auxiliary energy sources; the above equipment and facilities need to be used in conjunction with each other during operation. In the actual operation of the painting team, the equipment hardly ever stops, so only considering the workshop operation state can be divided into three states: processing, adjustment, and standby. During the operation of the painting workshop equipment, carbon emission and energy consumption show different situations. The specific situations are analyzed below.

- Manufacturing carbon emissions. Manufacturing carbon emissions represent the carbon emissions emitted by equipment within ship painting workshops during the operation. The overall manufacturing carbon emissions can be calculated as $C_W$ in Equation (1).

$$C_W = \sum_{i \in I} \sum_{j \in J} \sum_{m \in M_i} \frac{c_{i,j,m} \times p_{i,j} \times P_i^W \times \delta}{v_{m,i}} \tag{1}$$

- Adjustment of carbon emissions. Adjustment carbon emission refers to the carbon emission generated during the adjustment time required for the workshop equipment to process different segments successively. The total carbon emission under adjustment is calculated as $C_S$ shown in Equation (2).

$$C_S = \sum_{i \in I} \sum_{j \in J} \sum_{m \in M_i} \sum_{j' \in J} \sum_{q \in \{1,\ldots,n-1\}} x_{i,j,m,q} \times x_{i,j',m,q+1} \times T_{j,j',i}^S \times \delta \tag{2}$$

- Idle carbon emissions. Idle carbon emission refers to the carbon emission generated by the idle state of workshop equipment, and the total carbon emission under the idle state is calculated as $C_I$ shown in Equation (3).

$$C_I = \sum_{i \in I} \sum_{m \in M_i} \sum_{q \in \{1,\ldots,n-1\}} \left( B_{i,m,q+1} - E_{i,m,q} - \sum_{j \in J} \sum_{j' \in J} x_{i,j,m,q} \times x_{i,j',m,q+1} \times t_{j,j',i} \right) \times P_i^I \times \delta \tag{3}$$

- Auxiliary carbon emissions. Ancillary carbon emissions encompass the energy utilized during the transportation of segments, so the total transportation carbon emissions are calculated as $C_C$ shown in Equation (4).

$$C_C = \sum_{i \in [1,\ldots,s-1]} \sum_{j \in J} \sum_{m \in M_i} \sum_{m' \in M_{i+1}} \sum_{q \in L} \sum_{q' \in L} f_{m.m',i} \times x_{i,j,m,q} \times x_{i,j,m',q} \times z_{i,j} \times P_i^T \times \delta \quad (4)$$

### 2.4. Mathematical Model

Given the diverse carbon emissions from transportation and paint shop operations in different states, the development of a low-carbon scheduling model for ship painting minimizes completion time and total carbon emissions, as shown in Equations (5) and (6).

$$ob1 = MinC_{\max} \quad (5)$$

$$ob2 = MinEC = C_W + C_S + C_I + C_C \quad (6)$$

Two intermediate variables, $Y_{i,m,q}$ and $W_{i,j}$, are introduced to linearize it, and the linearized objective function is specified by Equation (7).

$$MinEC = \sum_{i \in I} \sum_{j \in J} \sum_{m \in M_i} \frac{c_{i,j,m} \times p_{i,j} \times P_i^W \times \delta}{v_{m,i}} + \sum_{i \in I} \sum_{m \in M_i} \sum_{q \in L} Y_{i,m,q} + \sum_{i \in [1,\ldots,s-1]} \sum_{j \in J} W_{i,j} \quad (7)$$

In addition, to minimize both objective functions at the same time, a linear weighting method is used to combine them into a single objective, where $w_1$ and $w_2$ are the weighting coefficients in Equation (8).

$$Min \ w_1 \times C_{\max} + w_2 \times EC, \quad (8)$$

which is subject to the following:

$$\sum_{m \in M_i} \sum_{q \in L} x_{i,j,m,q} = 1, \ \forall i \in I, j \in J \quad (9)$$

$$\sum_{j \in J} x_{i,j,m,q} \leq 1, \forall i \in I, m \in M_i, q \in L \quad (10)$$

$$\sum_{j \in J} x_{i,j,m,q} \geq \sum_{j \in J} x_{i,j,m,q+1} \ \forall i \in I, m \in M_i, q \in \{1,\ldots,n-1\} \quad (11)$$

$$e_{i,j} = b_{i,j} + p_{i,j}, \ \forall i \in I, j \in J \quad (12)$$

$$E_{i,m,q} = B_{i,m,q} + \sum_{j \in J} x_{i,j,m,q} \times p_{i,j}, \ \forall i \in I, m \in M_i, q \in L \quad (13)$$

$$e_{i,j} + \sum_{m \in M_i} \sum_{q \in L} f_{m,m',i} \times x_{i,j,m,q} \times z_{i,j} \leq b_{i+1,j} + \theta \times \left(1 - \sum_{q \in L} x_{i+1,j,m',q}\right), \ \forall i \in \{1,\ldots,s-1\}$$
$$j \in J, m' \in M_{i+1} \quad (14)$$

$$B_{i,m,1} \geq t_{j,j,i} - \theta \times (1 - x_{i,j,m,1}), \ \forall i \in I, j \in J, m \in M_i \quad (15)$$

$$\sum_{j \in J} t_{j,j',i} \times x_{i,j,m,q} + E_{i,m,q} \leq B_{i,m,q+1} + \theta \times \left(1 - x_{i,j',m,q+1}\right), \ \forall i \in I, \forall j' \in J$$
$$m \in M_i, q \in \{1,\ldots,n-1\} \quad (16)$$

$$B_{i,m,q} \leq b_{i,j} + \theta \times (1 - x_{i,j,m,q}), \ \forall i \in I, j \in J, m \in M_i, q \in L \quad (17)$$

$$B_{i,m,q} \geq b_{i,j} - \theta \times (1 - x_{i,j,m,q}), \ \forall i \in I, j \in J, m \in M_i, q \in L \quad (18)$$

$$Y_{i,m,1} \geq T_{j,j',i}^S - \theta \times (1 - x_{i,j,m,1}), \ \forall i \in I, j \in J, m \in M_i \quad (19)$$

$$\left( B_{i,m,q} - E_{i,m,q-1} - \sum_{j \in J} t_{j,j',i} \times x_{i,j,m,q-1} \right) \times P_i^I + \sum_{j \in J} T_{j,j',i}^S \times x_{i,j,m,q-1} \leq Y_{i,m,q} + \theta \times \left( 1 - x_{i,j',m,q} \right) \quad (20)$$
$$\forall j' \in J, i \in I, m \in M_i, q \in \{2, \ldots, n\}$$

$$\sum_{m \in M_i} \sum_{q \in L} x_{i,j,m,q} \times f_{m,m',i} \times p_i^F \times z_{i,j} \leq W_{i,j} + \theta \times \left( 1 - \sum_{q \in L} x_{i+1,j,m',q} \right) \quad (21)$$
$$\forall i \in \{1, \ldots, s-1\}, j \in J, m' \in M_{i+1}$$

$$b_{j,m} + \sum_{m \in M_s} \sum_{q \in L} p_{j,s} \times x_{j,s,m,q} \leq C_{\max}, \ \forall j \in J \quad (22)$$

$$0 \leq Y_{i,m,q}, \ \forall i \in I, m \in M_i, q \in L \quad (23)$$

$$b_{i,j}, B_{i,m,q} \geq 0, \ \forall i \in I, j \in J, m \in M_i, q \in L \quad (24)$$

$$x_{i,j,m,q} \in \{0,1\}, \ \forall i \in I, j \in J, m \in M, q \in L \quad (25)$$

$$z_{i,j} \in \{0,1\}, \ \forall i \in \{1, \ldots, s-1\}, j \in J \quad (26)$$

The segments processed by the painting team are defined by Equations (9)–(11). The allocation of each segment to a position within a painting team at stage *i* is specified by Equation (9). Only one segment can be assigned to each position of the paint team, as defined by Equation (10). According to Equation (11), the initial assignment of segments occurs at the front position of the painting team. The painting team processing segment constraints are defined by Equations (12)–(15). Defined by Equations (12) to (13), uninterrupted processing by the painting team during segment processing is ensured. At a specific stage, segment processing begins only upon the completion of processing in the sandblasting workshop and its transportation to the painting workshop via a flatbed truck, as defined by Equation (14). Equation (15) ensures that the first segment assigned to a segmental painting team begins processing only after adjustments have been made. Equation (16) ensures that segments (other than the first job) begin processing only after the segment at the previous location is completed and adjustments are completed for the segment at the later location. The linkage between $B_{i,m,q}$ and $b_{i,j}$ is specified by Equations (17) to (18). The definition of a variable $Y_{i,m,q}$ is outlined by Equations (19) through (20), while the definition of a variable $W_{i,j}$ is provided by Equation (21). Equation (22) defines the objective function $C_{\max}$. The remaining variables are defined by Equations (23)–(26).

*2.5. Illustrative Example*

The proposed low-carbon scheduling model for ship segmental painting results in different carbon emissions due to the different energy utilization rates of different painting groups in a given stage. If we prioritize lower carbon emissions, the work may be biased towards painting groups with higher energy utilization, while painting groups with lower energy utilization may be idle, which obviously reduces productivity and prolongs completion time. To illustrate this issue more clearly, the two stages where the main energy use is concentrated in a ship's segmented painting workshop, i.e., sandblasting and painting, are taken as examples. Below are the relevant production data.

$$p_{i,j} = \begin{bmatrix} 10 & 10 \\ 10 & 20 \\ 30 & 20 \\ 20 & 10 \end{bmatrix} f_{m,m',1} = \begin{bmatrix} 1 & 1 \\ 1 & 1 \end{bmatrix} v_{m,i} = \begin{bmatrix} 0.8 & 0.6 \\ 0.6 & 0.8 \end{bmatrix}$$

$$p_i^W = \begin{bmatrix} 6 & 5 \end{bmatrix} p_i^I = \begin{bmatrix} 1 & 1 \end{bmatrix} p_i^F = 2$$

$$t_{j,j',1} = \begin{bmatrix} 5 & 10 & 5 & 10 \\ 15 & 10 & 10 & 5 \\ 10 & 5 & 5 & 5 \\ 10 & 20 & 10 & 10 \end{bmatrix} t_{j,j',2} = \begin{bmatrix} 10 & 5 & 5 & 5 \\ 10 & 5 & 5 & 5 \\ 10 & 5 & 10 & 5 \\ 5 & 10 & 5 & 5 \end{bmatrix}$$

$$T_{j,j',1}^S = \begin{bmatrix} 10 & 20 & 20 & 10 \\ 30 & 20 & 10 & 20 \\ 20 & 10 & 10 & 10 \\ 10 & 20 & 10 & 10 \end{bmatrix} T_{j,j',2}^S = \begin{bmatrix} 20 & 20 & 10 & 10 \\ 20 & 10 & 10 & 10 \\ 20 & 10 & 10 & 20 \\ 10 & 20 & 10 & 10 \end{bmatrix}$$

The EC and MS of the ship segmental painting low-carbon scheduling workshop model using CPLEX12.1 are assigned weight vectors of size 0 to 1 with uniform distribution. It can be seen from Figure 2 that the total carbon emission of the painting workshop decreases with the increase in the maximum completion time, which indicates that EC and MS cannot be optimized at the same time; in addition, we can get the best EC and MS in different periods, so the relationship between these two objectives can be explained.

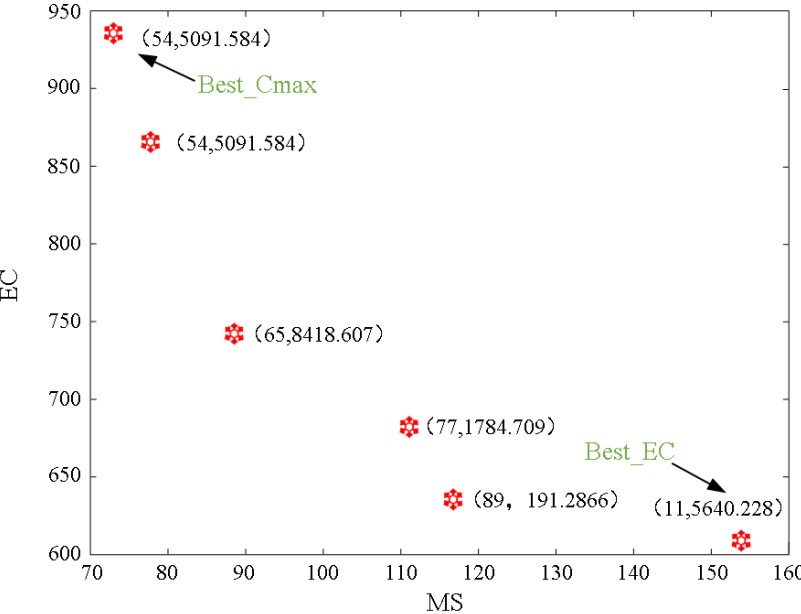

**Figure 2.** Completion time versus carbon emissions for the segmented painting workshop.

To illustrate the total carbon emission calculation, the carbon emission factor is taken as an example of scheduling in Figure 3, where the white rectangles indicate the adjustment time, the dashed rectangles indicate the transportation time, and the colored rectangles indicate the processing time. In the first stage of painting team 2, the total processing time is 30 + 10 = 40, and the processing carbon emission is 40 × 6 × 0.7559 ÷ 0.6 = 302.36. The adjustment carbon emission is (10 + 20) × 0.7559 = 22.677. The idle time is 0, and its carbon emission is also 0. Similarly, the carbon emission generated by the processing of the painting team in the other stages can be calculated. In addition, the carbon emission for transportation is 8 × 2 × 0.7559 = 12.0944. Thus, the final total carbon emission is 980.4023.

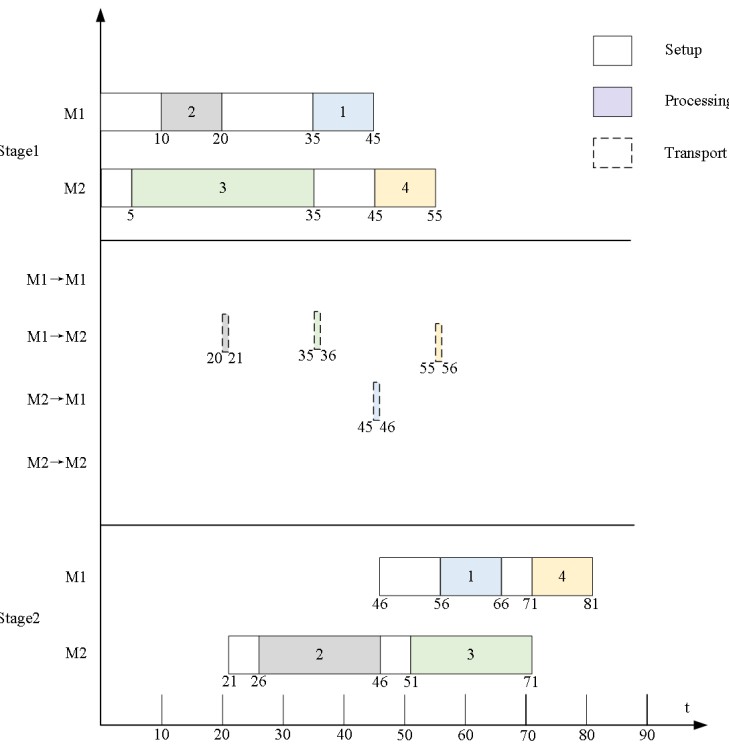

**Figure 3.** Low-carbon dispatch Gantt chart for ship segmental painting.

## 3. Basic Strategy

### 3.1. Decomposition Strategy

MOEA/D is one of the main representatives of decomposition methods, where each subproblem is defined by a weight vector, and an aggregation function is used to transform the MOP into multiple scalar problems that will be optimized during the population evolution process [26,27]. The aggregation function in this study uses the Tchebycheff method (TCH), defined in the below Equation (27):

$$g^{te}(x|w,z^*) = \max_{1 \le i \le m}\left\{\left|f(x) - z_i^*\right|/w^i\right\}, \tag{27}$$

where $x$ denotes the objective value of the corresponding subproblem of $w^i$, and $w^i$ denotes the weight vector corresponding to the $i$th subproblem and satisfies $\sum_{i=0}^{m} w^i = 1$; over here, it is important to note that $w^i$ cannot take the value of 0 in the denominator, but it can be taken as 0.00001. In addition, $z^* = (z_1^*, \ldots, z_m^*)$ denotes the ideal reference point, which in this study, is used as a reference point by defining the lower limit of the objective value as below.

### 3.2. Strategy of Angle-Based Selection

The same scalarization method used is TCH, and it can be applied to both convex and non-convex regions. In MOEA/D, weights are assigned to subproblems to naturally balance diversity and convergence. The precondition is for each subproblem to produce decentralized localized Pareto-optimal solutions. However, some cases may not conform to this assumption, and Figure 4 presents a case where A is defined by $w^1$. A plays a crucial role in diversity; however, it is likely to be displaced from B and C by other individuals. One problem with the current replacement strategy is that it is based on the choice of scalar values. This strategy suggests that if the aggregation function value of the offspring solution is equal to or lower than that of the parent solution, the offspring may replace the parent. To address this issue, a new selection strategy, called the strategy of angle-based selection, is introduced in this paper to retain diverse solutions [28–30].

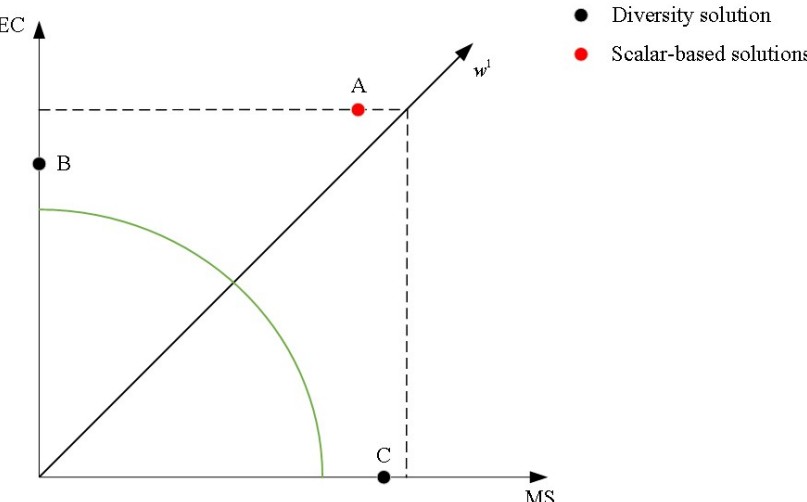

**Figure 4.** Schematic diagram of the dual-objective spatial solution.

Equation (28) initially establishes the acute angle formed by $w^i$ and $x^k$, where $w^{i_1}, \ldots w^{i_T}$ and $x^{i_1}, \ldots x^{i_T}$ are the weight vectors and solutions of the $T$ neighborhoods of $w^i$, respectively, and sets $B(i) = \{i_1, \ldots, i_T\}$.

$$angle(i,k) = \arccos\left(\frac{(F(x^k) - z^*)^T w^i}{||F(x^k) - z^*||||w^i||}\right) \tag{28}$$

Next, define the maximum angle $\varphi_i$ among the weight vectors $w^i$ and $x^k (k = 1, \ldots, N)$. If in the angle $angle(i,k)$ of candidate solutions $x^j$ and $w^j$, each subproblem is less than or equal to $\varphi_i$, then $x^j$ is a candidate solution for the parent individual and may be replaced. As shown in Figure 5 below, the vector $w^i$ has three adjacent weight vectors, $(w^{i_1}, w^{i_2}, w^{i_3})$, where $i_1 = i$. Points A, B, and C represent the three individuals of these three weight vectors, respectively. The maximum angle between the point and the weight vectors is $\varphi_i$. The area delineated by the two red dashed lines signifies the improved region. The core idea of the strategy of angle-based selection is to replace the old individuals using a selection strategy based on angles and scalarized values.

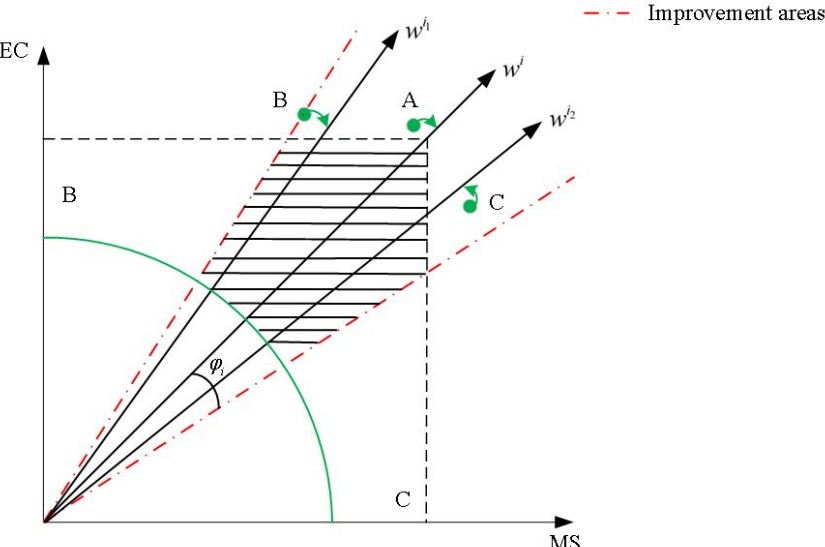

**Figure 5.** Improved regions based on angle selection strategies.

*3.3. Objective Normalization*

For MOPs with different target dimensions, the search may be biased towards targets with larger scales when the dimensionality difference is large. To address this issue, the introduction of goal normalization can greatly enhance the performance of algorithms employing a decomposition strategy, which is shown in Equation (29) below:

$$f_i\prime = \frac{f_i - \min(f_i)}{\max(f_i) - \min(f_i)},\tag{29}$$

where $f_i'$ is the target normalized value, $\max(f_i)$ denotes the upper bound of the target $f_i$ and $\min(f_i)$ denotes the bottom bound of that target. Each target value is normalized to [0,1]. In goal normalization, the requirement to ascertain true upper and lower bounds is eliminated [31]. Therefore, only their approximations need to be found, and a few definitions for certain extreme scenarios are provided here.

**Definition 1.** *Assume that each segmental painting workshop is handled only by the workshop painting team consecutively between any two phases such that $C_j^{min}(C_j^{max})$ denotes the completion time for handling segment j at the shortest (longest) adjustment time and the shortest (longest) conveyance time.*

$$\max(C_{\max}) = \sum_{j \in J} C_j^{\max}\tag{30}$$

$$\min(C_{\max}) = \max_{j \in J}\{C_j^{\min}\}\tag{31}$$

**Definition 2.** *Assume that all segments are assigned to be processed consecutively by the painting team with the largest (smallest) energy usage, and let $EC_{i,j}^l(EC_{i,j}^h)$ denote the total carbon emissions at the processing segment j with the smallest (largest) adjustment energy consumption and the smallest (largest) transportation time. When calculating $EC_{i,j}^l$, the idle time of the painting team is assumed to be zero, and when calculating $EC_{i,j}^h$, the idle time of the painting team is assumed to be the sum of the processing time and the adjustment time of the painting team in the previous stage.*

$$\max(EC) = \sum_{i \in I} \sum_{j \in J} EC_{i,j}^h\tag{32}$$

$$\min(EC) = \sum_{i \in I} \sum_{j \in J} EC_{i,j}^l\tag{33}$$

## 4. Decomposition-Based Multi-Objective Artificial Bee Colony Algorithm

This section presents an MD/ABC algorithm for solving the low-carbon scheduling of ship segmental painting. The MD/ABC algorithm aims to introduce the basic strategy described above into the framework of an efficient multi-objective artificial bee colony algorithm. The basic framework of the ABC algorithm is followed by a description of its coding and decoding scheme, a detailed explanation of the MD/ABC algorithm, and finally, the implementation of the algorithm.

*4.1. ABC Algorithm Basic Framework*

The solution to the problem addressed by the ABC algorithm consists of a nectar source, a lead bee, a follower bee, and a scout bee. Bees in different roles and holding different positions cooperate in the optimization process. The process of bees in search of high-quality nectar is the search for the optimal solution, and its basic framework is shown in Algorithm 1 [32,33].

| **Algorithm 1.** Fundamental structure of ABC algorithm |
|---|
| **Run Procedure** |
| Initialize (); |
|   **While** Not Termination Condition() **do** |
|     Employed Bee searching for nectar (); |
|     Onlooker Bee to choose the honey source (); |
|     Scout Bee look for new nectar sources in their neighborhoods (); |
|     **End While** |
|     Deliver the finest solution (); |
| **End Procedure** |

### 4.2. Initializing Populations

First, a random population of N solutions is generated, and the respective objective values are obtained through a decoding process. Then, the method proposed in Section 3.3 is used for normalization, and the solutions of the subproblems are evaluated by means of an aggregation function. Following the initialization of the population, an outer population of non-dominated solutions that maintain the population should be formed.

To obtain a uniformly distributed weight vector, the weight vector $w^i$ of the ith subproblem can be obtained from the set $\{0/H, 1/H, \ldots, H/H\}$, where $H$ is a control parameter that determines the population size. Thus, $N = C_{H+m-1}^{m-1}$, where $m$ is the target number, is here set to 2, and the value of $H$ is equal to $N-1$.

### 4.3. Encoding and Decoding

One of the core issues in algorithm design is the encoding of the solution, and a good encoding method should cover the comprehensive solution space information and ensure the search efficiency of the algorithm. In the ship segmental painting low-carbon scheduling problem, the objectives are MS and EC. To achieve the optimization of these two objectives, it is necessary to determine the order of the segments in each stage and the allocation of the segments of each painting team in each stage so that the solution is encoded in two layers. The first layer is an n-dimensional arrangement, $\pi_n = \{\pi_1, \ldots, \pi_j, \ldots, \pi_n\}$, where $\pi_j$ denotes the segment index and $n$ denotes the number of segments. The segments that appear first in the ranking have higher priority to be processed in the first stage. The second layer is the painting team selection matrix $v_n$, where $v_{i,j}$ denotes the painting team assigned to segment $j$ on process $i$.

$$v_{i,j} = \begin{bmatrix} v_{1,1} & v_{1,2} & \ldots & v_{1,n} \\ \ldots & & & \ldots \\ v_{i,1} & & & v_{i,2} \end{bmatrix}$$

To decode the solution, into a feasible scheduling, the detailed decoding process is summarized below. In the first stage, for the paint group, 0 moments are available. Take out the segments $\pi_j$ one by one, according to $\pi_n$, and perform the following steps.

Step 1: Find the segmental painting team assigned to the segment and its available time in the matrix $v_{i,j}$.

Step 2: Calculate the completion time for the segment $\pi_i$ and then update the available time for the specified segmental painting team.

Step 3: Find the assigned painting team $v_{2,\pi_i}$ in the next stage and then calculate the start time $\pi(i)$ by adding the transportation time.

Subsequent phases are to first ensure that the segmentation sequence $\pi_n' = \{\pi'(1), \pi'(2), \ldots, \pi'(n)\}$ has taken $\pi'(i)$ from $\pi_n'$ one by one according to their respective start times, and perform the following steps.

Step 1: Find the segmental painting team $v_{m,\pi'(i)}$ assigned by the segmental painting and its available time in the matrix $v_{i,j}$.

Step 2: Calculate the completion time for the segment $\pi'(i)$ and then update the available time for the specified segmental painting team.

Step 3: Find the assigned painting team $v_{m+1,\pi\prime(i)}$ in the next stage and then calculate the start time $\pi\prime(i)$ by adding the transportation time.

### 4.4. Changing Neighborhoods to Lead the Employed Bee Phase

In the employed bee phase, the employed bee improves itself to be able to fully explore the environment around the solution. In the employed bee phase, the employed bee improves itself to be able to fully explore the environment around the solution. Solution-based coding has two coupling layers, each corresponding to a different problem and neighborhood structure, which are switched by designing five methods of switching neighborhood structures to allow for the global exploration of the solution space.

- Segment insertion: randomly select segments from the segmented sequence vector and insert them into different randomly selected segments;
- Segment swap: randomly select two segments from the segmented sequence vector and swap their positions;
- Paint group mutation: randomly select segments from the paint group assignment matrix and change their assigned paint groups to different paint groups.
- Perform segmental painting insertion followed by painting panel mutation.
- Perform segmental swap followed by painting group mutation.

To make full use of the current neighborhood structure, it is designed to switch between the five neighborhoods to explore the solution space. Suppose $X_i$ denotes the solution of the *i*th subproblem, and $S_i(S_i \in [1,\dots,5])$ denotes the current neighborhood structure index. Algorithm 2 shows the procedure of the employed bee phase. Here, $Neighborhood_{S_i}(X_i)$ denotes obtaining the neighboring solution $X_i$ by manipulating the neighborhood $S_i$ for $X'_i$, and $UpdateEP(X'_i)$ denotes updating the external population using $X'_i$. $R_i$ denotes the number of consecutive failed updates of the current solution, and if the maximum number of consecutive failed updates of the current solution is greater than $C$, then the current neighborhood structure is switched to the next neighborhood structure.

---

**Algorithm 2.** Changing neighborhoods to lead the employed bee phase

---

**For** $i = 1$ to $N$ **do**
  $X'_i \leftarrow Neighborhood_{s_i}(X_i)$;
  $UpdateEP(X'_i)$;
    **If** $g(X'_i|w^i) < g(X_i|w^i)$ **then**
      $X_i \leftarrow X'_i; R_i \leftarrow 0; S_i \leftarrow 1$;
    **ELse**
      $X_i \leftarrow X_i$ ; $R_i \leftarrow R_i + 1$ ; $S_i \leftarrow S_i + 1$;
    **Endif**
    **If** $R_i > C$ **then**
      $S_i \leftarrow S_i + 1$;
    **Endif**
    **If** $S_i > 5$ **then**
      $S_i \leftarrow 1$;
      **EndIf**
**Endfor**

---

### 4.5. Collaborative Onlooker Bee Phase

In the phase, depending on the quality of the nectar source provided by the employed bee, the onlooker bee selects the nectar source with more potential. To expedite algorithm convergence and achieve superior-quality solutions, employ the TOPSIS and the binary tournament approaches [34]. Two solutions are randomly chosen. Their similarity to the ideal solution is measured according to TOPSIS. The solution with the higher value is chosen. This method operates on the principle that an effective solution should be near the ideal solution and distant from the perspective of the negative ideal solution. Similarity to the ideal solution can be calculated using Equation (34).

$$S_i^+ = \frac{d_i^-}{d_i^+ + d_i^-},$$

(34)

where $S_i^+$ represents the similarity between the *i*th solution and the ideal solution, and $d_i^-$ and $d_i^+$ denote the Euclidean distances of the *i*th solution from the negative ideal and ideal of the target space, respectively. In this paper, the upper and lower bounds of these two objectives constitute the ideal and negative solution objectives, respectively. Initially, two distinct individuals are chosen and evaluated using TOPSIS. Then, if the superior individual follows the inferior one, their positions are swapped; otherwise, they are left unaltered. To enhance the probability of positioning a promising individual at the front of each line, the competitive process is repeated ten times after each tournament. The use of this mechanism ensures that promising solutions have more chances to be utilized.

Promising solutions at this stage use crossover operators that allow subproblems to collaborate with other subproblems [35]. In this paper, the dual-site crossover (TPX) process using the genetic operator for segmental painting sequences and the matrix selection operator for painting groups is shown in Figures 6 and 7 below to simulate the honeybee information-sharing process, which results in an offspring that inherits the characteristics of both parents and is used to update the biparental solution if the offspring has a better fitness.

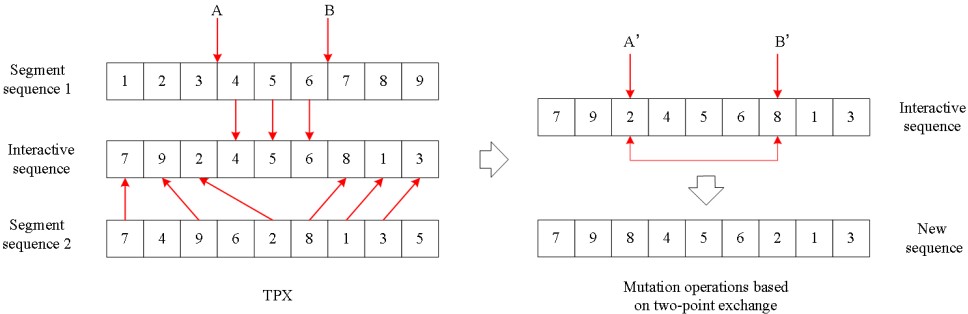

**Figure 6.** Genetic operators for segmented sequences.

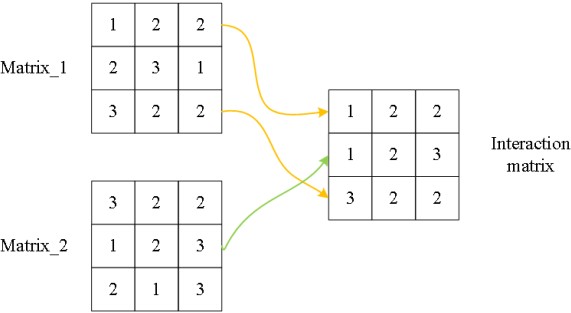

**Figure 7.** Selection operators for paint Group matrices.

To perform TPX, the solution partners should be chosen, and a neighborhood relationship between subproblems should be established. Due to the proximity of solutions in neighboring subproblems to each other, there is a probability of producing more promising offspring. The Euclidean distance is used to compute the distance between subproblems from which the nearest T subproblems are selected to form a neighborhood. Next, a neighboring subproblem is randomly chosen, and its solution is selected as a partner. Nonetheless, depending solely on weight vectors to measure proximity can lead to solutions becoming similar, consequently reducing population diversity. To solve this problem, an angle-based selection strategy is introduced. Specific steps are shown in Section 3.2. The whole process of the onlooker bee phase is shown in Algorithm 3, where *Select*() denotes the operator that uses TOPSIS-based binary tournament rules to select a good solution $X_A$, $FPS(X_A)$ denotes the operator that finds a partner solution $X_A$ to the solution $X_B$ in the subproblem neighborhood using an angle-based selection strategy, and $TPX()$ denotes the two-point crossover operator.

**Algorithm 3.** Collaborative onlooker bee phase

**For** $i = 1$ to $N$ **do**
$X_A \leftarrow Select()$;
$\quad\quad X_B \leftarrow FPS(X_B)$;
$X_{new} \leftarrow TPX(X_A, X_B)$
$Update(X_{new})$;
$\quad\quad\quad$ **If** $rand < \delta$ **then**
$\quad\quad\quad\quad\quad\quad E = B(i)$;
$\quad\quad\quad$ **else**
$\quad\quad\quad\quad\quad\quad E = A$;
$\quad\quad\quad$ Randomly select two distinct indexes r1,r2 that are different from I from E;
$\quad\quad\quad$ Generate the offspring y by the Algorithm 2;
$\quad\quad\quad$ Evaluate **y** and update $z^*$;
$\quad\quad\quad$ Compute the angles between $w^j (j \in E)$ and **y** denoted as $\varphi_{new}$ using(28);
$\quad\quad\quad$ $\varphi_{old} = \varphi(E)$;
$\quad\quad\quad$ $c = 0$;
$\quad\quad\quad$ **while** $(c < n_r)$ **&** (E is not null) **do**
$\quad\quad\quad\quad\quad\quad$ Randomly pick an index j from E;
$\quad\quad\quad\quad\quad\quad$ $\gamma = \frac{g^{\text{tch}}(x^j|w^j,z^*) - g^{\text{tch}}(y|w^j,z^*)}{g^{\text{tch}}(x^j|w^j,z^*)}$;
$\quad\quad\quad\quad\quad\quad$ $\delta = \varphi_{old}(j) - \varphi_{new}(j)$;
$\quad\quad\quad\quad\quad\quad$ **if** $\gamma \geq 0$ **&** $\delta \geq 0$ **then**
$\quad\quad\quad\quad\quad\quad\quad\quad$ Replace $x^j$ with y, and set $c = c + 1$;
$\quad\quad\quad\quad\quad\quad\quad\quad$ Remove j from E;
$\quad\quad\quad\quad\quad\quad$ **end**
$\quad\quad$ **end**

### 4.6. Solving the Exchange of the Scout Bee Phase

When a solution is not updated within a long period of time, it is considered abandoned, its corresponding employed bee is remodeled into a scout bee, and a new solution is randomly selected. Since the randomized strategy is not under control, it does not have a positive impact on the efficiency of the algorithm. To improve the efficiency of the algorithm in this phase, a neighborhood-based solution exchange strategy is used. This is because the similarity of solutions between neighboring subproblems does not affect their evolution. Suppose that $X_i^k$ represents the solution of the $T$th neighbor subproblem of the $i$th subproblem, where $k \in [1, 2, \ldots, T]$. Algorithm 4 presents the pseudo-code procedure for the scout bee phase, where $L(X_i)$ denotes the $L$ value of $X_i$ obtained.

**Algorithm 4.** Solving the exchange of the employed bee phase

**For** $i = 1$ to $N$ **do**
$\quad$ **If** $L(X_i) > L$ **then**
$\quad\quad$ $k \leftarrow 1$
$\quad\quad$ num is exchanged $\leftarrow$ false;
$\quad$ **While** is exchange=false **then**
$\quad\quad$ **If** $X_i > X_i^k$ **then**
$\quad\quad\quad$ $X_i \leftrightarrow X_i^k$
$\quad\quad\quad$ is exchange=true;
$\quad\quad$ **Else**
$\quad\quad\quad\quad\quad$ $k++$;
$\quad\quad$ **End**
$\quad\quad$ **If** $k > T$ **then**
$\quad\quad\quad\quad$ $r = Rand(1, T)$;
$\quad\quad\quad\quad$ $X_i \leftarrow X_i^r$;
$\quad\quad\quad\quad$ Is Exchanged $\leftarrow$ true;
$\quad\quad$ **End**
$\quad$ **End**
**End**

### 4.7. The Complete Algorithmic Process

This section summarizes the whole process of the MD/ABC algorithm, starting with the input parameters.

$N$: the total of subproblems.
$w^i$ ($i = 1, \ldots, N$): uniformly distributed weight vectors.
$T$: number of neighboring subproblems in a neighborhood.
$C$: maximum number of consecutive failed updates for the current solution.
$L$: number of consecutive failed iterations before discarding the solution.

The algorithm then works as follows:

Step 1: Initialize the population, which contains N randomly generated solutions and N distributed weight vectors described in Section 4.2.

Step 2: Update the population and the external population.

Step 2.1: Perform the neighborhood-based switching method described in Section 4.4 applied to the employed bee phase.

Step 2.2: Execute the angle selection strategy described in Section 4.5 applied to the collaborative onlooker bee phase.

Step 2.3: Execute the solution-based switching described in Section 4.6 applied to the scout bee phase.

Step 3: Repeat Step 2 until the termination condition is satisfied.

Step 4: Update the external population and output.

The flow of the MD/ABC algorithm is shown in Figure 8:

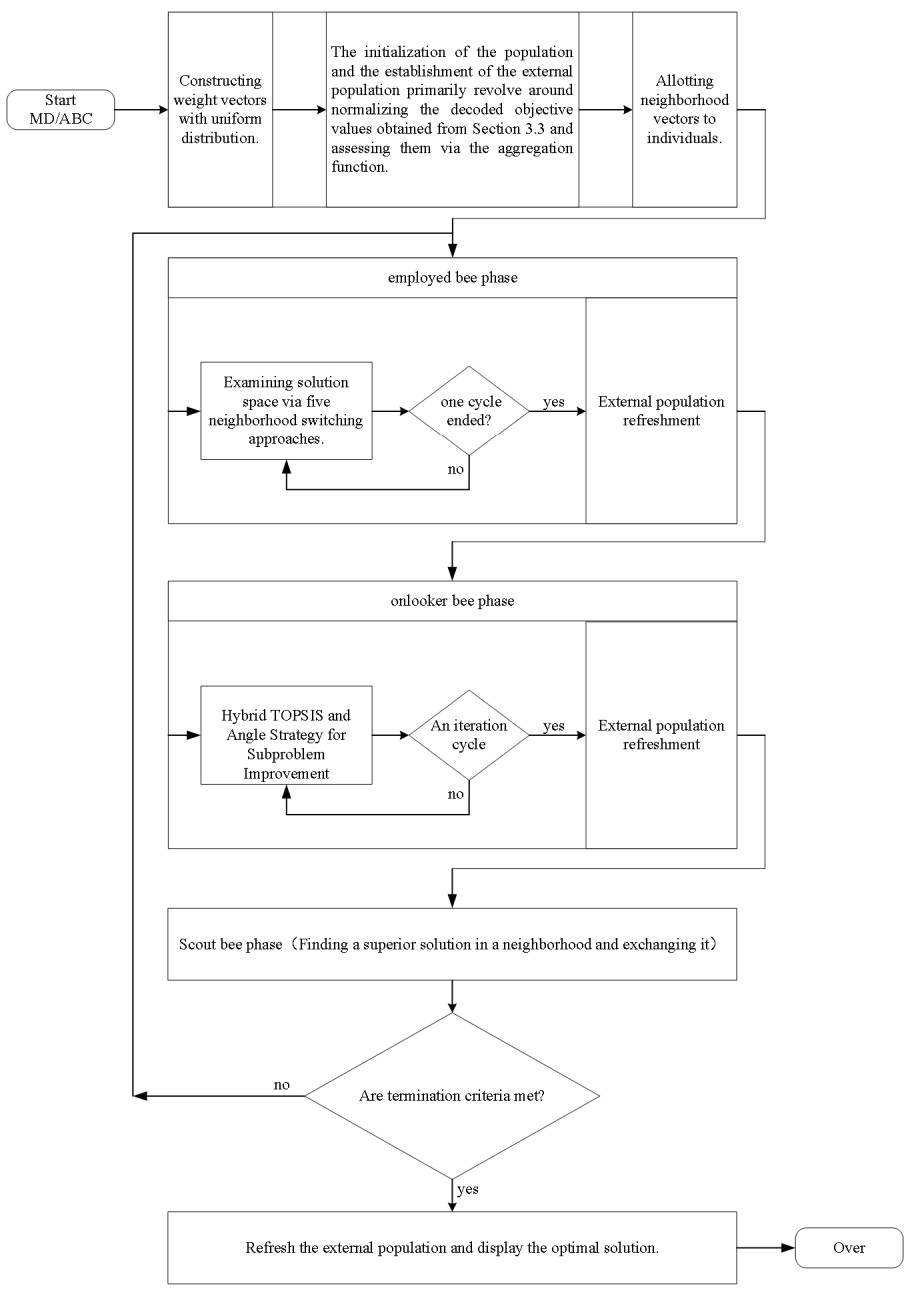

**Figure 8.** MD/ABC complete algorithm flow.

## 5. Testing Study

The purpose of this section is to perform numerical experiments on the MD/ABC algorithm and to evaluate the algorithm and the proposed strategy. Firstly, the parameter settings of the low-carbon scheduling of ship segmental painting are analyzed, and the performance metrics of the algorithms are evaluated. Then, the proposed strategies are compared to other strategies through comparative experiments. Finally, a comparison experiment between MD/ABC and other algorithms is conducted. In the MD/ABC algorithm, the termination criterion is set to n × m × μ milliseconds, where n represents the number of segments, m represents the number of stages, and in comprehensive experiments, μ is set as a fixed value of 200. For all compared algorithms in the experiments, C++ coding is employed, and all experiments are conducted on an Intel Pentium processor running at 3.10 GHz.

*5.1. Test Data*

Static processing data are generated based on the low-carbon scheduling configuration for ship segmental painting described in Part I. The detailed process of data generation is described below. It is widely used in HFSPs with sequence-dependent setup times. To comprehensively assess the MD/ABC algorithm's performance from various angles, different problem instances need to be selected for testing. The parameters *n*, *m*, and *lv* control the problem instances. The processing times for each segment $n \in \{20, 40, 60, 80, 100\}$ and stage $m \in \{3, 5, 8, 10\}$ are randomly sampled from a uniform distribution $U[1, 99]$, while the number of painting teams per stage is uniformly distributed as $U[1, 5]$. The durations for transportation between particular stages are sampled uniformly from the distribution $U[1, 25]$. The number of stages is determined by the number of segments N and the number of stages M processing times. There are four levels of sequence-dependent setup times, which are drawn from the uniform distributions $U[1, 25]$, $U[1, 49]$, $U[1, 99]$, and $U[1, 124]$. The setup times acquired correspond to 25%, 50%, 100%, and 125% of the average processing time. Thus, by combining *n*, *m*, and *lv*, 80 problem configurations can be obtained. Each configuration requires the generation of five instances, amounting to a combined total of 400 instances. In addition, carbon emission and energy consumption information need to be added to extend the above instances, and the total carbon emission is the product of the total energy consumption in each state of the shop and the carbon emission factor, which is taken as 0.7559 kg $CO_2$/kWh in this paper. The handling power is generated from $pp_i = 4 \times c_i$, where the $c_i$ factor is taken from the real interval $U[1.0, 2.0]$. The adjusted power $sp_i$ is 2, the idle power is 1, the unloaded and transported power are both 1, and the energy utilization is sampled from the uniform distribution $U[0.7, 1.0]$. In summary, Table 2 lists the test data used to generate the low-carbon scheduling of painting operations on ship segments.

**Table 2.** Summary of test data.

| Factors | Levels | Number of Levels |
|---|---|---|
| Number of segments | 20, 40, 60, 80, 100 | 5 |
| Number of stages | 3, 5, 8, 10 | 4 |
| Number of painting teams per stage | $U[1,5]$ | 1 |
| Transportation time | $U[1,25]$ | 1 |
| Basic processing time | $U[1,99]$ | 1 |
| Conversion factor | $U[1.0, 2.0]$ | 1 |
| Adjustment time | $U[1,25]$, $U[1,49]$, $U[1,99]$, $U[1,124]$ | 4 |
| Adjustment energy consumption | $U[2,5]$ | 1 |
| Idle energy consumption | $U[1,3]$ | 1 |

*5.2. Performance Indicators*

This paper adopts four key performance metrics to comprehensively evaluate the performance of the MD/ABC algorithm [36–38].

- Spread is a distributional metric that measures the Pareto frontier (*PF*), reflecting the degree of coverage of the solution set, with higher values indicating greater uniformity and better diversity, which is defined as follows:

$$S(A, PF^*) = \frac{\sum_{j=1}^{m} d_j^e + \sum_{i=1}^{|A|} \left| d_i - \bar{d} \right|}{\sum_{j=1}^{m} d_j^e + \left| A \right| \cdot \bar{d}} \tag{35}$$

  *A* denotes the resulting *PF*, while *PF*\* representing the true solution set. $d_i$ represents the shortest Euclidean distance between the current solution *i* and the real solution set, and the average Euclidean distance *A* is denoted by $\bar{d}$. $|A|$ and *m* signify the total number of members and objectives, respectively. $d_j^e$ represents the Euclidean distance from objective *j* to the extreme solutions of the actual solution. Concerning this metric, smaller values are better.

- Generational distance (GD) represents the divergence between the acquired *PF* and the real *PF*, which measures the quality of convergence of the obtained *PF*, which is defined as follows:

$$GD(A, PF^*) = \frac{\sqrt{\sum_{v \in A} d(v, PF^*)^2}}{|A|} \tag{36}$$

- The Inverse Generation Distance (IGD) is a comprehensive metric, and it can be used to evaluate the distance and distribution between the solution set and the ideal solution, which is defined as follows:

$$IGD(A, PF^*) = \frac{\sum_{V \in PF^*} d(v, A)}{|PF^*|} \tag{37}$$

  Here, $d(v, A)$ stands for the minimum Euclidean distance between *v* and the points within *A*, with $|PF^*|$ indicating the count of points in *PF*\*.

- The number of nondominated solutions (NOS) is used to indicate the number of non-dominated solutions in the approximated true frontier, with larger values providing a better approximation of the entire true frontier.

### 5.3. Parameter Settings

Initialization parameters include the number of subproblems in the population N, the number of neighboring subproblems for each subproblem (T), the number of neighboring solutions generated (M), the maximum number of consecutive update failures for solutions using the current neighborhood structure (C), and the number of consecutive failures to abandon a solution in successive generations (L). In MD/ABC, there are five parameters containing N, T, M, C, and L. This ensures that the five different neighborhood structures are fully utilized under the selection strategy and scouting mechanism. To gain empirical insights into the impact of these five parameters and to set them efficiently, the Taguchi method is used to apply examples [39]. Table 3 lists four plausible levels for the five parameters, with their combinations determined by the L16 orthogonal array listed in Table 4. Each combination undergoes independent execution of the MD/ABC algorithm 30 times, where the average IGD metric (AVG) is collected as the response variable and is shown in Table 4. Based on the factor level trends shown in Table 4 and presented in Figure 9, for each parameter, the significance rankings are listed in Table 5. As can be seen from Table 4 and Figure 9, larger N values can promote exploratory capabilities but contradict the goal of deep search of subproblems under the finite termination criterion. The parameter T has a significant effect, and the algorithm performs best when T = 20, whereas too small a value may promote collaboration of very similar solutions, which may lead to insufficient global exploration, whereas too large a value may lead to a waste of computational resources. From Figure 9, it can be seen that the performance of the algorithm gets progressively worse as the value of M increases; this is because when

M = 2, individuals make optimal use of neighboring solutions. As the value of M increases, more computational resources will be consumed, thus decreasing the performance of the algorithm. The results obtained through designed experiments show that the algorithm performs best when N = 150, M = 2, T = 20, C = 10, and L = 50.

**Table 3.** Level of the parameters.

| Parameter | Parameter Level | | | |
|---|---|---|---|---|
| | 1 | 2 | 3 | 4 |
| N | 100 | 150 | 200 | 250 |
| M | 2 | 3 | 5 | 8 |
| T | 5 | 10 | 15 | 20 |
| C | 8 | 10 | 15 | 18 |
| L | 20 | 30 | 50 | 80 |

**Table 4.** The orthogonal array L16.

| Experiment Number | Parameter | | | | | AVG |
|---|---|---|---|---|---|---|
| | N | M | T | C | L | |
| 1 | 100 | 2 | 5 | 8 | 20 | 0.0904 |
| 2 | 100 | 3 | 10 | 10 | 30 | 0.0922 |
| 3 | 100 | 5 | 15 | 15 | 50 | 0.0873 |
| 4 | 100 | 8 | 20 | 18 | 80 | 0.0762 |
| 5 | 150 | 2 | 10 | 15 | 80 | 0.0785 |
| 6 | 150 | 3 | 5 | 18 | 50 | 0.0931 |
| 7 | 150 | 5 | 20 | 8 | 30 | 0.0792 |
| 8 | 150 | 8 | 15 | 10 | 20 | 0.0901 |
| 9 | 200 | 2 | 15 | 18 | 30 | 0.0882 |
| 10 | 200 | 3 | 20 | 15 | 20 | 0.0742 |
| 11 | 200 | 5 | 5 | 10 | 80 | 0.0942 |
| 12 | 200 | 8 | 10 | 8 | 50 | 0.0933 |
| 13 | 250 | 2 | 20 | 10 | 50 | 0.0543 |
| 14 | 250 | 3 | 15 | 8 | 80 | 0.0921 |
| 15 | 250 | 5 | 10 | 18 | 20 | 0.1255 |
| 16 | 250 | 8 | 5 | 15 | 30 | 0.1121 |

**Table 5.** Ranking of significance levels.

| Level | N | M | T | C | L |
|---|---|---|---|---|---|
| 1 | 0.0865 | 0.0779 | 0.0975 | 0.0886 | 0.0951 |
| 2 | 0.0852 | 0.0879 | 0.0974 | 0.0827 | 0.0929 |
| 3 | 0.0875 | 0.0966 | 0.0894 | 0.0880 | 0.0820 |
| 4 | 0.0960 | 0.0929 | 0.0710 | 0.0958 | 0.0853 |
| Rank | 4 | 3 | 2 | 5 | 1 |

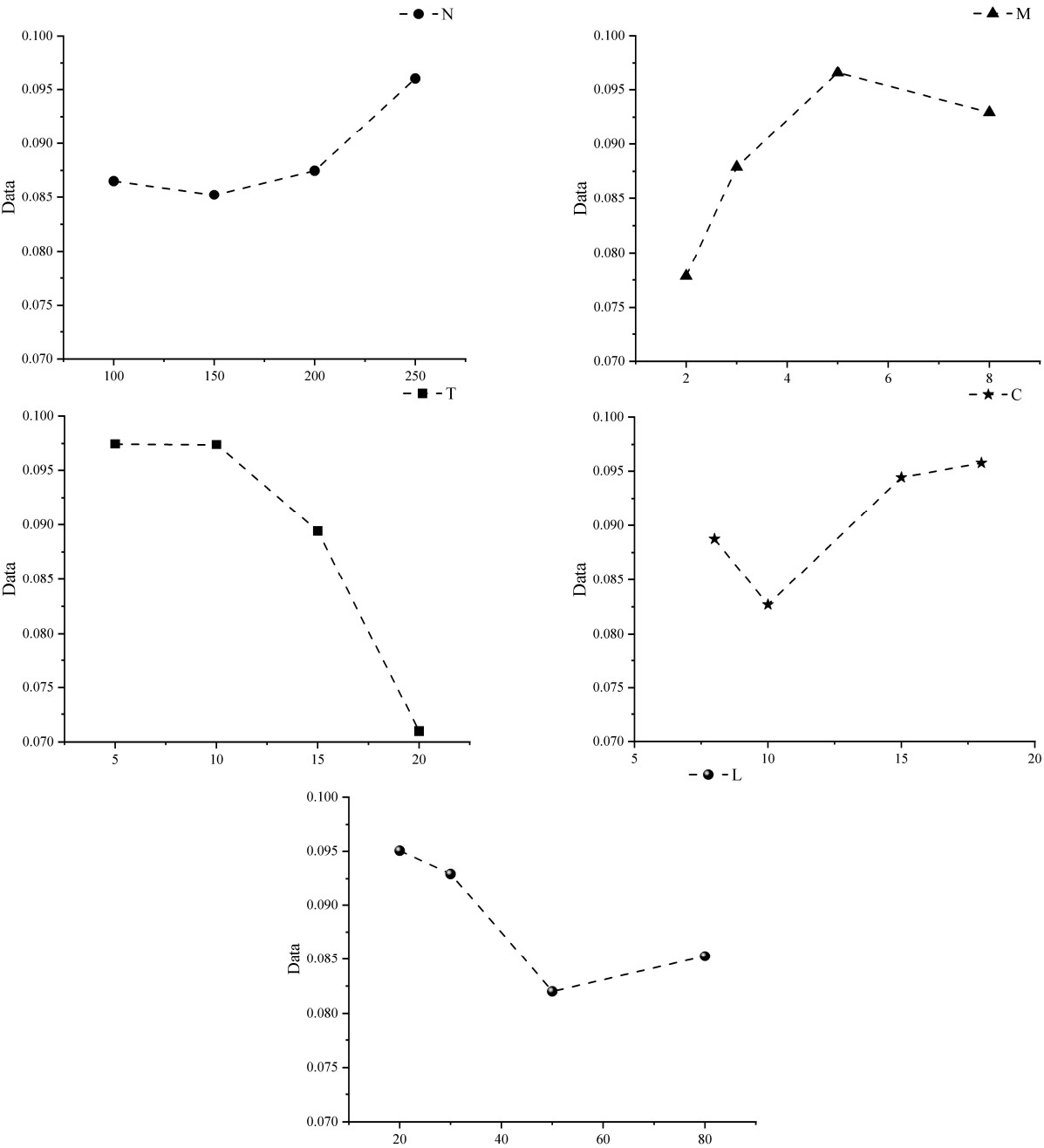

**Figure 9.** Parameter level trends.

## 5.4. Evaluate the Core Strategy

The purpose of this section is to evaluate the effectiveness of several core strategies, including the angle-based selection strategy and the solution exchange strategy. MD/ABCa does not employ an angle-based selection strategy, while MD/ABCs uses a randomized strategy instead of a solution exchange strategy. For a fair comparison, all parameter settings of MD/ABCa and MD/ABCs are the same as MD/ABC. To demonstrate their combined performance, for each of the 400 instances, the average value (AGV) of the IGDs is obtained by 30 independently repeated runs, and they are categorized by the identical problem size and averaged once more. Table 6 gives the results of the comparison, presenting the overall IGD values and highlighting the best values for each problem in bold at the end.

**Table 6.** Comparative results of the variants of MD/ABC on test instances regarding IGD.

| Problem | MD/ABC | MD/ABCa | MD/ABCs |
|---|---|---|---|
| 20 × 3 | 0.074 | **0.072** | 0.124 |
| 20 × 5 | 0.086 | **0.081** | 0.153 |
| 20 × 8 | **0.095** | 0.145 | 0.192 |
| 20 × 10 | 0.142 | **0.136** | 0.188 |
| 40 × 3 | 0.082 | **0.076** | 0.143 |
| 40 × 5 | **0.144** | 0.212 | 0.234 |
| 40 × 8 | **0.096** | 0.098 | 0.165 |
| 40 × 10 | 0.181 | **0.178** | 0.242 |
| 60 × 3 | 0.165 | **0.162** | 0.231 |
| 60 × 5 | **0.114** | 0.132 | 0.199 |
| 60 × 8 | 0.144 | **0.139** | 0.233 |
| 60 × 10 | **0.111** | 0.163 | 0.187 |
| 80 × 3 | **0.101** | 0.104 | 0.206 |
| 80 × 5 | 0.103 | **0.101** | 0.190 |
| 80 × 8 | **0.122** | 0.221 | 0.304 |
| 80 × 10 | **0.163** | 0.236 | 0.258 |
| 100 × 3 | 0.082 | **0.078** | 0.178 |
| 100 × 5 | 0.186 | **0.146** | 0.286 |
| 100 × 8 | **0.083** | 0.086 | 0.158 |
| 100 × 10 | 0.121 | **0.118** | 0.257 |
| Mean | **0.120** | 0.134 | 0.206 |

The best value for each question is marked in bold.

As can be seen in Table 6, MD/ABC has the lowest overall IGD mean value (0.120) compared with MD/ABCa and MD/ABCs. Specifically, the algorithm performs best on 9 out of 20 test instances compared with MD/ABCa, while MD/ABCa outperforms MD/ABC on 11 test instances. This is mainly because the angle-based selection strategy restricts the region of improvement, which leads to the weakening of convergence, and the diversity decreases as the number of iterations in the target space increases. For MD/ABC, the distribution of individuals can be maintained well and eventually approaches the real *PF* gradually. Compared with MD/ABCs, MD/ABC wins in all problems, which is because the solution exchange strategy can utilize evolutionary information and enhance search efficiency. These two evaluation strategies highlight the ability of global exploration and local exploitation, respectively. MD/ABCs excel in the overall metrics; therefore, it can be concluded that the balance between global and local exploitation ability can be achieved by adopting the perspective-based selection strategy and the solution exchange strategy, which can be verified.

*5.5. Evaluating the MD/ABC Algorithm*

This section aims to evaluate the performance of the MD/ABC algorithm by comparing it to other algorithms [40–44]. These algorithms include NSGAII, NSGAIII, MOEA/D, iMOEA/D, EAMOA, and MOABCD. iMOEA/D and EAMOA are used to solve energy-efficient HFSP problems, while NSGAII, NSGAIII, and MOEA/D are the three most popular algorithms for solving a wide range of optimization problems. nSGAII and NSGAIII use a fast nondominated sorting strategy, while MOEA/D uses a decomposition strategy. iMOEA/D is an improved version of MOEA/D for solving combinatorial optimization problems by a strategy that compensates for the efficiency problem of the decomposition strategy. MOABCD is a multi-objective variant of the ABC algorithm for solving combinatorial optimization problems, also based on a decomposition strategy.

Both NSGAII and NSGAIII frameworks are similar, yet they employ different selection mechanisms. NSGAII ranks solutions by their crowding distances, while NSGAIII incorporates widely distributed reference points for selection. All compared algorithms use Taguchi's method to set the parameters appropriately. The encoding and decoding of the solutions are kept consistent with MD/ABC, and specific operators are adapted

to the problem. The performance of the algorithms is fully tested and based on Spread, GD, IGD, and NOS metrics. The averaged results were subjected to a one-way analysis of variance (ANOVA) to assess their statistical validity. When the confidence intervals of two algorithms overlap, it implies no statistically significant difference between them. The results of the comparison are presented in Tables 7–10, and the ANOVA plots are displayed in Figures 10–12, respectively.

**Table 7.** Comparison AVG based on Spread.

| Problem | MD/ABC | NSGAII | NSGAIII | EAMOA | MOEA/D | iMOEA/D | MOABCD |
|---|---|---|---|---|---|---|---|
| 20 × 3 | **0.725** | 1.178 | 1.362 | 0.954 | 0.864 | 0.754 | 0.815 |
| 20 × 5 | **0.517** | 1.122 | 1.143 | 0.966 | 0.673 | 0.709 | 0.743 |
| 20 × 8 | **0.613** | 1.083 | 1.113 | 0.975 | 0.726 | 0.694 | 0.740 |
| 20 × 10 | **0.611** | 1.058 | 1.024 | 0.954 | 0.984 | 0.752 | 0.819 |
| 40 × 3 | **0.629** | 1.162 | 1.216 | 0.932 | 0.762 | 0.769 | 0.796 |
| 40 × 5 | **0.617** | 1.167 | 1.243 | 0.965 | 0.843 | 0.748 | 0.738 |
| 40 × 8 | **0.612** | 1.147 | 1.257 | 0.944 | 0.720 | 0.753 | 0.836 |
| 40 × 10 | **0.681** | 1.095 | 1.087 | 0.939 | 0.735 | 0.691 | 0.726 |
| 60 × 3 | **0.621** | 1.106 | 1.098 | 0.812 | 0.697 | 0.736 | 0.744 |
| 60 × 5 | **0.701** | 1.136 | 1.145 | 0.923 | 0.802 | 0.707 | 0.812 |
| 60 × 8 | **0.642** | 1.105 | 1.101 | 0.824 | 0.817 | 0.662 | 0.772 |
| 60 × 10 | **0.741** | 1.082 | 1.017 | 0.912 | 0.846 | 0.795 | 0.783 |
| 80 × 3 | **0.015** | 1.225 | 1.125 | 0.924 | 0.811 | 0.654 | 0.785 |
| 80 × 5 | **0.547** | 1.071 | 1.165 | 0.905 | 0.898 | 0.745 | 0.753 |
| 80 × 8 | **0.124** | 1.067 | 1.154 | 0.956 | 0.983 | 0.681 | 0.703 |
| 80 × 10 | **0.499** | 1.074 | 1.098 | 0.912 | 0.978 | 0.758 | 0.762 |
| 100 × 3 | **0.712** | 1.212 | 1.232 | 0.976 | 0.801 | 0.736 | 0.728 |
| 100 × 5 | **0.649** | 1.078 | 1.178 | 0.934 | 0.905 | 0.690 | 0.764 |
| 100 × 8 | **0.641** | 1.129 | 1.178 | 0.859 | 0.816 | 0.656 | 0.879 |
| 100 × 10 | **0.578** | 1.102 | 1.095 | 0.992 | 0.992 | 0.723 | 0.898 |
| Mean | **0.574** | 1.120 | 1.152 | 0.928 | 0.833 | 0.721 | 0.780 |

The best value for each question is marked in bold.

**Table 8.** Comparison AVG based on GD.

| Problem | MD/ABC | NSGAII | NSGAIII | EAMOA | MOEA/D | iMOEA/D | MOABCD |
|---|---|---|---|---|---|---|---|
| 20 × 3 | **0.014** | 0.028 | 0.039 | 0.017 | 0.044 | 0.022 | 0.021 |
| 20 × 5 | **0.022** | 0.042 | 0.052 | 0.034 | 0.053 | 0.041 | 0.037 |
| 20 × 8 | **0.026** | 0.035 | 0.048 | 0.031 | 0.051 | 0.038 | 0.034 |
| 20 × 10 | **0.012** | 0.037 | 0.034 | 0.030 | 0.050 | 0.033 | 0.026 |
| 40 × 3 | **0.016** | 0.049 | 0.036 | 0.027 | 0.036 | 0.018 | 0.041 |
| 40 × 5 | **0.027** | 0.034 | 0.040 | 0.033 | 0.039 | 0.042 | 0.038 |
| 40 × 8 | **0.023** | 0.036 | 0.057 | 0.035 | 0.037 | 0.038 | 0.032 |
| 40 × 10 | 0.029 | 0.044 | 0.031 | 0.046 | 0.044 | **0.026** | 0.038 |
| 60 × 3 | **0.027** | 0.054 | 0.048 | 0.039 | 0.041 | 0.031 | 0.045 |
| 60 × 5 | **0.024** | 0.048 | 0.042 | 0.037 | 0.049 | 0.029 | 0.039 |
| 60 × 8 | **0.026** | 0.031 | 0.037 | 0.034 | 0.042 | 0.032 | 0.029 |
| 60 × 10 | **0.020** | 0.045 | 0.032 | 0.044 | 0.051 | 0.035 | 0.026 |
| 80 × 3 | **0.016** | 0.039 | 0.046 | 0.031 | 0.037 | 0.019 | 0.031 |
| 80 × 5 | **0.019** | 0.041 | 0.048 | 0.038 | 0.031 | 0.028 | 0.035 |
| 80 × 8 | **0.022** | 0.058 | 0.051 | 0.037 | 0.035 | 0.025 | 0.040 |
| 80 × 10 | **0.025** | 0.036 | 0.046 | 0.031 | 0.042 | 0.028 | 0.046 |
| 100 × 3 | **0.022** | 0.042 | 0.031 | 0.041 | 0.038 | 0.026 | 0.044 |
| 100 × 5 | **0.018** | 0.041 | 0.055 | 0.031 | 0.044 | 0.056 | 0.033 |
| 100 × 8 | **0.027** | 0.036 | 0.047 | 0.039 | 0.042 | 0.034 | 0.032 |
| 100 × 10 | **0.015** | 0.039 | 0.038 | 0.030 | 0.034 | 0.058 | 0.035 |
| Mean | **0.022** | 0.041 | 0.043 | 0.034 | 0.042 | 0.033 | 0.035 |

The best value for each question is marked in bold.

**Table 9.** Comparison AVG based on IGD.

| Problem | MD/ABC | NSGAII | NSGAIII | EAMOA | MOEA/D | iMOEA/D | MOABCD |
|---|---|---|---|---|---|---|---|
| $20 \times 3$ | **0.064** | 0.088 | 0.097 | 0.074 | 0.076 | 0.074 | 0.079 |
| $20 \times 5$ | **0.045** | 0.117 | 0.108 | 0.068 | 0.071 | 0.082 | 0.082 |
| $20 \times 8$ | **0.047** | 0.096 | 0.102 | 0.080 | 0.074 | 0.081 | 0.054 |
| $20 \times 10$ | **0.053** | 0.081 | 0.124 | 0.063 | 0.069 | 0.076 | 0.067 |
| $40 \times 3$ | **0.037** | 0.072 | 0.094 | 0.074 | 0.075 | 0.079 | 0.052 |
| $40 \times 5$ | **0.063** | 0.076 | 0.106 | 0.072 | 0.068 | 0.073 | 0.068 |
| $40 \times 8$ | **0.067** | 0.080 | 0.086 | 0.076 | 0.079 | 0.078 | 0.075 |
| $40 \times 10$ | **0.072** | 0.085 | 0.079 | 0.082 | 0.087 | 0.080 | 0.071 |
| $60 \times 3$ | **0.062** | 0.087 | 0.079 | 0.071 | 0.083 | 0.083 | 0.077 |
| $60 \times 5$ | **0.048** | 0.063 | 0.082 | 0.075 | 0.084 | 0.078 | 0.061 |
| $60 \times 8$ | **0.058** | 0.078 | 0.081 | 0.079 | 0.088 | 0.061 | 0.065 |
| $60 \times 10$ | **0.063** | 0.087 | 0.086 | 0.068 | 0.085 | 0.076 | 0.070 |
| $80 \times 3$ | **0.057** | 0.083 | 0.103 | 0.072 | 0.076 | 0.077 | 0.074 |
| $80 \times 5$ | **0.072** | 0.088 | 0.087 | 0.077 | 0.086 | 0.071 | 0.088 |
| $80 \times 8$ | **0.054** | 0.082 | 0.099 | 0.081 | 0.078 | 0.075 | 0.076 |
| $80 \times 10$ | **0.071** | 0.086 | 0.086 | 0.084 | 0.086 | 0.074 | 0.082 |
| $100 \times 3$ | **0.061** | 0.074 | 0.089 | 0.074 | 0.072 | 0.076 | 0.085 |
| $100 \times 5$ | **0.072** | 0.077 | 0.076 | 0.073 | 0.074 | 0.084 | 0.074 |
| $100 \times 8$ | **0.062** | 0.089 | 0.091 | 0.083 | 0.080 | 0.075 | 0.086 |
| $100 \times 10$ | **0.061** | 0.083 | 0.093 | 0.082 | 0.077 | 0.085 | 0.081 |
| Mean | **0.059** | 0.084 | 0.092 | 0.075 | 0.078 | 0.077 | 0.073 |

The best value for each question is marked in bold.

**Table 10.** Comparison AVG based on NOS.

| Problem | MD/ABC | NSGAII | NSGAIII | EAMOA | MOEA/D | iMOEA/D | MOABCD |
|---|---|---|---|---|---|---|---|
| $20 \times 3$ | **63** | 55 | 47 | 42 | 42 | 46 | 6 |
| $20 \times 5$ | **79** | 34 | 22 | 47 | 27 | 31 | 11 |
| $20 \times 8$ | **43** | 21 | 36 | 16 | 33 | 27 | 8 |
| $20 \times 10$ | **52** | 26 | 38 | 8 | 50 | 36 | 17 |
| $40 \times 3$ | **92** | 16 | 21 | 23 | 81 | 82 | 32 |
| $40 \times 5$ | **51** | 12 | 8 | 16 | 37 | 31 | 16 |
| $40 \times 8$ | **42** | 9 | 11 | 9 | 34 | 40 | 14 |
| $40 \times 10$ | **49** | 6 | 19 | 7 | 34 | 36 | 23 |
| $60 \times 3$ | **87** | 39 | 32 | 25 | 72 | 74 | 26 |
| $60 \times 5$ | **71** | 17 | 24 | 8 | 69 | 59 | 10 |
| $60 \times 8$ | **54** | 13 | 31 | 18 | 37 | 41 | 11 |
| $60 \times 10$ | **52** | 12 | 15 | 7 | 42 | 40 | 14 |
| $80 \times 3$ | **106** | 83 | 46 | 38 | 84 | 73 | 12 |
| $80 \times 5$ | **84** | 27 | 37 | 10 | 70 | 69 | 27 |
| $80 \times 8$ | **39** | 22 | 22 | 16 | 32 | 24 | 30 |
| $80 \times 10$ | **67** | 13 | 14 | 19 | 65 | 54 | 16 |
| $100 \times 3$ | **77** | 61 | 49 | 44 | 74 | 53 | 31 |
| $100 \times 5$ | **47** | 36 | 25 | 25 | 29 | 32 | 15 |
| $100 \times 8$ | **65** | 11 | 13 | 13 | 54 | 55 | 14 |
| $100 \times 10$ | **52** | 23 | 17 | 8 | 36 | 31 | 17 |
| Mean | **64** | 27 | 26 | 20 | 50 | 47 | 18 |

The best value for each question is marked in bold.

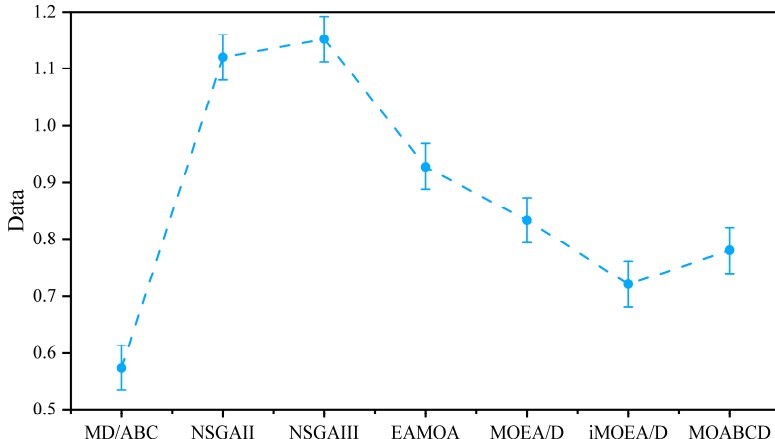

**Figure 10.** ANOVA-based effect plot on Spread.

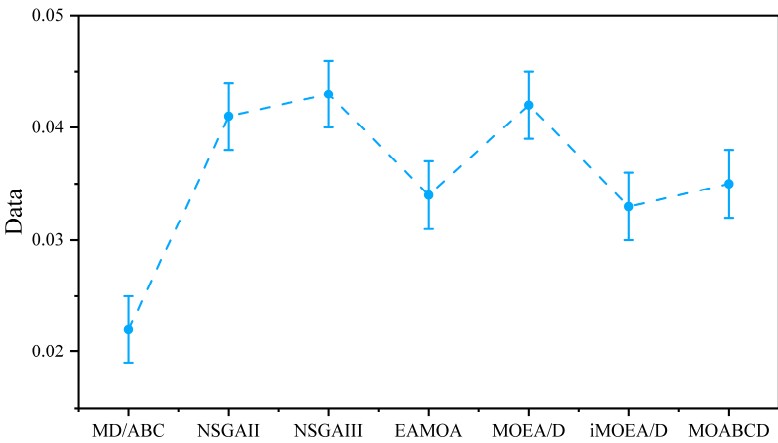

**Figure 11.** ANOVA-based effect plot on GD.

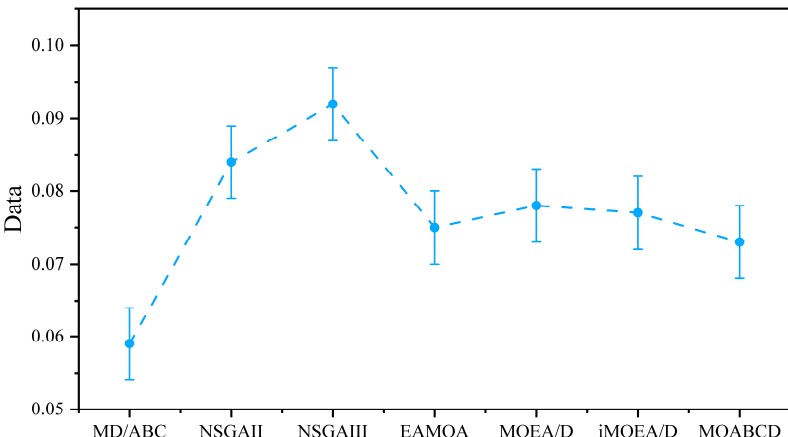

**Figure 12.** T ANOVA-based effect plot on IGD.

According to the Spread metrics in Table 7, MD/ABC performs the best among the 20 problems, showing its robustness and ability to generate uniformly distributed solutions. As seen from the ANOVA results in Figure 10, the MD/ABC algorithm significantly outperforms the other algorithms, which is attributed to its angle-based selection strategy and scouting mechanism, which helps to perform an adequate search along the neighboring weights to obtain a uniformly distributed Pareto solution. Furthermore, it is observed that algorithms using decomposition strategies, such as MD/ABC, MOEA/D, iMOEA/D, and

MOABCD, usually outperform NSGAII, NSGAIII, and EAMOA, prompting the adoption of decomposition strategies to solve the problem. According to the GD metrics in Table 8, MD/ABC performs best in 19 out of 20 problems, reaching the minimum overall value, indicating that it is capable of obtaining solutions closest to the true Pareto frontier. The advantage of the MD/ABC algorithm is more obvious as the problem size increases. Therefore, it can be concluded that MD/ABC has better global exploration ability and can jump out the local optimal solution efficiently. As can be seen from Figure 11, the MD/ABC algorithm is significantly better than the other algorithms. In terms of IGD metrics, it can be observed from Table 9 and Figure 12 that the MD/ABC algorithm again outperforms the other algorithms based on the superiority of the Spread and GD metrics, suggesting that MD/ABC has better diversity and convergence in terms of the quality of the solution set. In terms of NOS metrics, the NOS values are given in Table 10, with MD/ABC obtaining the maximum average overall problems, producing a larger overall average, supporting the previous conclusions. To visualize and demonstrate the performance of the algorithm, Figures 13 and 14 present the Pareto frontier distributions for the four instances (20 × 8, 40 × 8, 60 × 8, and 80 × 8) and the IGD value curves for convergence over time, respectively. From Figure 13, it can be seen that the solutions generated by MD/ABC are more uniformly distributed. Figure 14 shows that the convergence curve of MD/ABC is smoother, indicating that the algorithm is robust enough to obtain uniformly distributed solutions with good convergence. In summary, the effectiveness of MD/ABC in solving the low-carbon scheduling problem of ship segmental painting is proven.

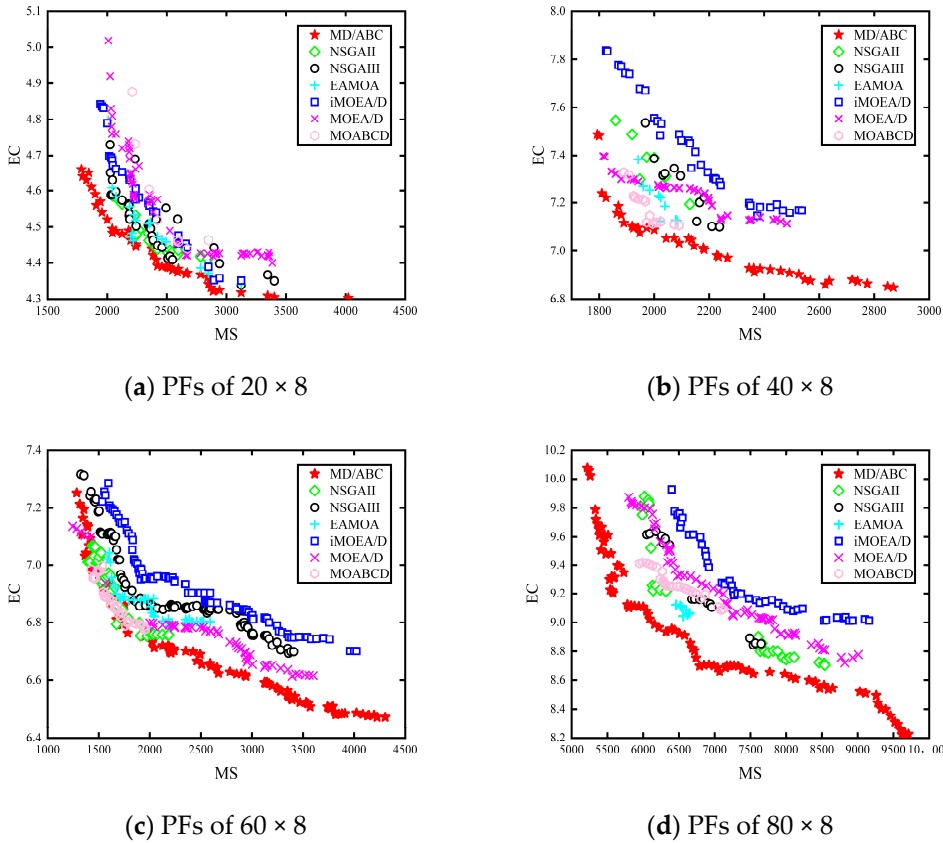

**Figure 13.** Pareto fronts of the instances obtained by the compared algorithms.

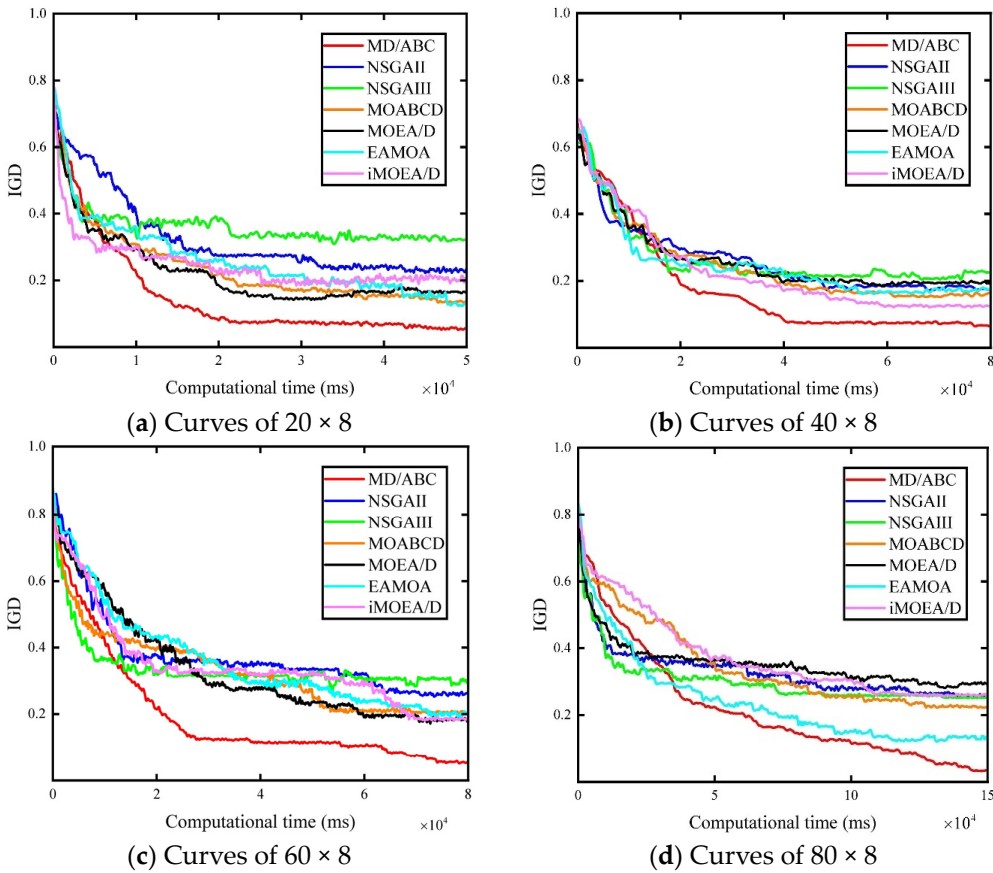

**Figure 14.** Convergence graphs for different scales.

## 6. Conclusions and Future Work

In this paper, to address the problems of low scheduling efficiency and excessive carbon emissions in the ship segmental painting scheduling process, a low-carbon scheduling model for ship segmental painting was established, and a multi-objective evolutionary algorithm MD/ABC based on the improvement of decomposition strategy was put forward. In the MD/ABC algorithm, a two-level coding method was designed to increase the quality of the solution. By designing five kinds of neighborhood switching methods, the method can fully explore the neighborhoods of the solution. The TOPSIS technique was used to improve the competition mechanism, while the angle selection strategy was introduced to further improve the diversity and convergence of the algorithm and realize the improvement of the ABC algorithm. In addition, Section 5.3 rationalized the settings of five parameters in the MD/ABC algorithm. From the trends in parameter levels and significance rankings, it is evident that the parameters T, M, and L have a significant impact on the experiments. T and M are related to neighborhood search; smaller values promote cooperation, while larger values lead to lower convergence. Appropriate settings can achieve a balance between global and local exploration. In relation to parameter L, smaller values tend to discard solutions that have not been fully explored, while larger values result in the wastage of computational resources. Section 5.4 further evaluated the effectiveness of angle selection strategies and solution exchange strategies by comparing several variants of the MD/ABC algorithm. Experimental results demonstrate that these mechanisms facilitate the cooperation between solutions and promote population co-evolution. In Section 5.5, based on performance evaluation indicators, a comparison of algorithms from different literature showed that the MD/ABC algorithm proposed in this paper outperforms others. Not only does it achieve a better balance between local search and global exploration, but it also enhances the diversity and convergence of the population. In conclusion, the MD/ABC algorithm performs excellently in reducing carbon emissions in painting work-

shop scheduling and can provide reliable guidance for scheduling schemes in shipyard segment painting workshops, with promising application prospects.

Due to the complexity and uncertainty of the ship segmental painting process, this paper did not consider too many constraints and objectives. In future work, we intend to consider more problem-specific features in segmental painting scheduling to improve the results and evaluate other objectives. We intend to consider the dynamics of shipbuilding projects and develop adaptive scheduling methods to address uncertainties in the painting process, such as repainting and expedited segment arrivals. Furthermore, at the algorithmic level, the scalability of the MD/ABC algorithm will be considered to make it suitable for large-scale ship segment painting scheduling problems involving more segments and painting teams.

**Author Contributions:** H.B. revised the paper and completed it; X.Z. wrote the first draft of the paper; Z.G., T.Y. and Y.T. collected and sorted the data; Z.Y. provided financial support. All authors have read and agreed to the published version of the manuscript.

**Funding:** The authors gratefully acknowledge the financial support from the Ministry of Industry and Information Technology High-Tech Ship Research Project: Research on the Development and Application of a Digital Process Design System for Ship Coating (No.: MC-202003-Z01-02), the National Defense Basic Scientific Research Project: Research and Development of an Intelligent Methanol-Fueled New Energy Ship (No.: JCKY2021414B011), and the RO-RO Passenger Ship Efficient Construction Process and Key Technology Research (No.: CJ07N20).

**Institutional Review Board Statement:** Not applicable.

**Informed Consent Statement:** Not applicable.

**Data Availability Statement:** The data that support the findings of this study are available from the corresponding author upon reasonable request.

**Conflicts of Interest:** The authors declare no conflicts of interest.

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
