# Peer review of "A Decomposition-Based Multi-Objective Evolutionary Algorithm for Solving Low-Carbon Scheduling of Ship Segment Painting"

_coatings, doi:10.3390/coatings14030368_

Round 1

Reviewer 1 Report

Comments and Suggestions for Authors

The introduction needs to be revised, summarizing the initial part, providing bibliographic references for the supported claims, and enhancing the state-of-the-art regarding the algorithms currently used internationally. Additionally, paragraph 2.1 Segmental painting process is unclear and fails to segue into paragraph 2.2. The introduction should outline the advantages and disadvantages of the different algorithms concerning the segments presented in paragraph 2.2.

Please provide a clearer explanation in section 2.5 regarding the choice of the example and the segments considered in reference to paragraph 2.2, which should be distinct from the introduction.

The results comprise a mathematical treatment founded on postulates only partially expounded in preceding paragraphs.

Author Response

Special thanks to you for your good comments. We have tried our best to improve the manuscript and made some changes in the manuscript. These changes will not influence the content and framework of the paper, and the changes we did not list here have been marked up using the “Track Changes” in revised paper. We appreciate Editors/Reviewers' warm work earnestly and hope the correction will meet with approval. Once again, thank you very much for your comments and suggestions.

Reviewer 2 Report

Comments and Suggestions for Authors

1. Several abbreviations were used without first writing them out in full with the abbreviations following in parentheses and afterwards the abbreviations can then be used in the manuscript. 

2. Authors cited Several works and narrated the outcomes of the paper's but did not filter out the gap(s) in them. The review of literature should be critically and analytically done and should be geared towards finding out gaps or pending issues in literature. 

3. Authors have not provided the research problem/gap that is borne from prior related works.  The study background is poorly written. 

4. The methodology and study protocols are not in a stepwise and systematic manner.  Authors should provide a step-by-step protocol for the study.  A diagrammatic representation may be helpful. 

5. The results presented were not discussed.  In fact, the discussion of findings is missing in the manuscript. 

6. Authors should indicate the how their findings compare and/or contrast the findings from prior related studies.

7. The implications and contributions of the study findings to practice should be indicated. 

8. What are the limitations of the study? Authors should indicate the threats to the validity of the experiments/study and indicate how the threats were mitigated against. 

9. Most of the cited references are old.  An upgrade is required. 

Author Response

(The authors gave the same response as above.)

Reviewer 3 Report

Comments and Suggestions for Authors

This paper introduces a novel approach to tackle the issue of excessive carbon emissions in ship segment painting. The authors propose a decomposition-based multi-objective artificial bee colony algorithm (MD/ABC) to optimize the painting process. The algorithm incorporates various techniques such as neighborhood switching methods, TOPSIS technique, and angle selection strategy to enhance local search, competition mechanism, and population diversity. Additionally, a two-stage coding method is designed specifically for the scheduling problem. The effectiveness of the MD/ABC algorithm is demonstrated through comprehensive tests using examples from a shipyard's segmented painting workshop. Although the paper is interesting, the following comments should be addressed.

-- To strengthen the introduction, it would be beneficial to include more specific details on the current challenges faced by existing scheduling methods in ship construction, particularly in the context of low-carbon painting.

-- It would be valuable to provide a more comprehensive and in-depth analysis of the existing literature specifically related to ship segment painting.

-- Also, discuss more recent publications including: ‘Solving multi-objective functions for cancer treatment by using metaheuristic algorithms’, ‘Factors affecting public transportation use during pandemic: an integrated approach of technology acceptance model and theory of planned behavior’, A machine learning approach to credit card customer segmentation for economic stability, and ‘Investigation of the impacts of the refill valve diameter on prestrike occurrence in gas’.

-- The authors should point out the major contributions of this paper by using 3 to 5 brief bullet points at the end of the Introduction section, right before the last paragraph.

-- The structure of arguments needs to be improved. At the end of the introduction part, you should have a section plan (for example section 2 discusses... and section 3 gives...).

-- It would be beneficial to provide more detailed explanations and justifications for the design decisions made in the algorithm, such as the selection of neighborhood switching methods and the TOPSIS technique.

-- It is recommended to present and discuss the performance metrics used for evaluation, such as completion time, carbon emissions reduction, and algorithm convergence.

-- I suggest considering the dynamic nature of ship construction projects and developing adaptive scheduling algorithms capable of handling uncertainties and changes in the painting process.

--It is recommended to investigate the economic aspects of low-carbon scheduling, such as cost optimization and resource allocation, to provide a comprehensive decision-making framework for the shipbuilding industry.

-- I suggest investigating the scalability of the MD/ABC algorithm for larger ship segment painting problems involving a higher number of segments and painting operators.

Comments on the Quality of English Language

Minor editing of the English language is required.

Author Response

(The authors gave the same response as above.)

Reviewer 4 Report

Comments and Suggestions for Authors

The article deals with the complex multilateral problem of painting a certain incomprehensible object. For example, the annotations indicate "The painting of ships (which?), as one of the three pillars of the shipping industry, goes through the entire process of building ships (for what conditions?). However, there are problems such as low planning efficiency, long completion time and large carbon dioxide emissions during the segment painting process" - the issue is solved one-sidedly and narrowly, does not take into account the specifics of the complexity and tasks of engineering design, this is the first, second after mathematical modeling, and when using a certain software model, the real object is not taken into account, it is not specified Maybe there are no specific recommendations for what? What are the natural conditions? Painting a new and old object? As well as recommendations for a narrow conclusion, the article does not meet the set goals, I cannot recommend the article for publication because of the general judgment about the model without reference to a specific object.

Author Response

(The authors gave the same response as above.)

Round 2

Reviewer 1 Report

Comments and Suggestions for Authors

The authors have change the manuscript and now it can be accepted 

Author Response

(The authors gave the same response as above.)

Reviewer 2 Report

Comments and Suggestions for Authors

1. Authors assert that "shipping companies are facing a green transition due to the huge increase in carbon emissions and energy consumption costs". Authors should provide support for ghost assertion. 

2. The results presented were either not discussed or not adequately discussed. 

a) Authors should indicate how their study findings agree and/or disagree with the findings from prior related studies. 

b) Authors should indicate the implications and contributions of the study findings to theory and practice. 

c) Insights should be provided for all findings of the study. 

3. The limitations of the study should be given and the directions for future studies should be based on these limitations. 

Author Response

(The authors gave the same response as above.)

Reviewer 3 Report

Comments and Suggestions for Authors

The authors have carefully addressed all the reviewing comments raised in the first round of review. It's my pleasure to recommend acceptance of the current version.

Author Response

(The authors gave the same response as above.)

Reviewer 4 Report

Comments and Suggestions for Authors

The question was answered positively, and the article became more understandable to the reader. Taking into account the importance of the direction of the stage and method of research and further promising directions for the development of the research topic, as well as taking into account significant positive changes in  conclusion, the expert recommends publications on the profile of the journal.

Author Response

(The authors gave the same response as above.)
